EMBO
Molecular Medicine

# Neutrophils suppress tumor-infiltrating T cells in colon cancer via matrix metalloproteinase-mediated activation of TGFβ

Markus Germann[1,*] (ID), Nadine Zangger[1,2,3], Marc-Olivier Sauvain[4,5], Christine Sempoux[6], Amber D Bowler[1], Pratyaksha Wirapati[2,3], Lana E Kandalaft[4,7], Mauro Delorenzi[2,3], Sabine Tejpar[8], George Coukos[4,7] & Freddy Radtke[1,**] (ID)

## Abstract

High T-cell infiltration in colorectal cancer (CRC) correlates with a favorable disease outcome and immunotherapy response. This, however, is only observed in a small subset of CRC patients. A better understanding of the factors influencing tumor T-cell responses in CRC could inspire novel therapeutic approaches to achieve broader immunotherapy responsiveness. Here, we investigated T cell-suppressive properties of different myeloid cell types in an inducible colon tumor mouse model. The most potent inhibitors of T-cell activity were tumor-infiltrating neutrophils. Gene expression analysis and combined *in vitro* and *in vivo* tests indicated that T-cell suppression is mediated by neutrophil-secreted metalloproteinase activation of latent TGFβ. CRC patient neutrophils similarly suppressed T cells via TGFβ *in vitro,* and public gene expression datasets suggested that T-cell activity is lowest in CRCs with combined neutrophil infiltration and TGFβ activation. Thus, the interaction of neutrophils with a TGFβ-rich tumor microenvironment may represent a conserved immunosuppressive mechanism in CRC.

**Keywords** colorectal cancer; neutrophils; T-cell suppression; TGF-β; tumor microenvironment
**Subject Categories** Cancer; Immunology

## Introduction

Inflammation is a major hallmark of cancer (Grivennikov *et al*, 2010; Hanahan & Weinberg, 2011), and both pro- and anti-tumorigenic properties of inflammatory cells have been described. T cell-mediated anti-tumor immune response correlates with favorable disease outcome and is the basis of cancer immunotherapy (Angell & Galon, 2013; Ribas & Wolchok, 2018). Many myeloid cells, such as macrophages, dendritic cells, monocytes, and neutrophils, not only act as antigen presenting cells to promote anti-tumor T-cell responses, but also suppress T cells in many cancer contexts (Grivennikov *et al*, 2010; Joyce & Fearon, 2015; Kumar *et al*, 2016). In CRC, immune cell infiltration and contexture are heterogeneous and reflective of its pathology (Guinney *et al*, 2015; Dienstmann & Tabernero, 2017; Pages *et al*, 2018). Activated T cells are most prominent in primary CRC tumors characterized by high microsatellite instability (MSI) of cancer cells (Mlecnik *et al*, 2016), and infiltration thereof is correlated with favorable prognoses (Pages *et al*, 2018). Remarkable success of immune checkpoint blockade (ICB), which bolsters T-cell tumor immunity, in solid tumors (Ribas & Wolchok, 2018), led to initiation of clinical trials for ICB in CRC. Unfortunately, the trials were largely unsuccessful and an effective response was only observed in MSI cases (Le *et al*, 2017; Overman *et al*, 2017). The vast unresponsiveness of CRC patients to ICB therapy suggests that there may be unknown factors contributing to T-cell suppression in the tumor microenvironment (TME).

Immune-promoting and immune-suppressive signals are present in normal colon and are essential to maintain tissue homeostasis (Saleh & Trinchieri, 2011). The lack of anti-tumor immune responses observed in a variety of cancers including CRC is thought to depend largely on an aberrant activation of immune-suppressive

---

1   Swiss Institute for Experimental Cancer Research (ISREC), School of Life Sciences Ecole Polytechnique Fédérale de Lausanne (EPFL), Lausanne, Switzerland
2   Bioinformatics Core Facility, Swiss Institute of Bioinformatics (SIB), Lausanne, Switzerland
3   Department of Oncology, Translational Bioinformatics and Statistics, Swiss Cancer Center Lausanne, University of Lausanne, Lausanne, Switzerland
4   Department of Oncology, Lausanne University Hospital, Lausanne, Switzerland
5   Department of Visceral Surgery, Lausanne University Hospital, Lausanne, Switzerland
6   Institute of Pathology, Lausanne University Hospital, Lausanne, Switzerland
7   Ludwig Institute for Cancer Research, Lausanne, Switzerland
8   Digestive Oncology Unit, University Hospital Gasthuisberg, Leuven, Belgium
    *Corresponding author. Tel: +41 79 4631922; E-mail: markus.germann@gmx.net
    **Corresponding author. Tel: +41 21 69330771; E-mail: freddy.radtke@epfl.ch

signals in the TME (Llosa *et al*, 2015; Topalian *et al*, 2015; Le *et al*, 2017; Overman *et al*, 2017). Whether these immune-evasive mechanisms are induced exclusively in established CRCs or develop first in early lesions is currently unknown. Similar to CRC, benign adenomatous precursor lesions are variably infiltrated with T cells and myeloid cells (McLean *et al*, 2011; Maglietta *et al*, 2016). How this correlates with and influences progression to CRC, however, is largely unknown.

Mouse models with inducible intestinal adenoma formation recapitulate human disease and allow for the functional investigation of the TME and its implication in tumor progression (Jackstadt & Sansom, 2016). Here, we investigate the influence of different subsets of immune cells on murine colon adenoma formation and progression. We demonstrate that tumor-infiltrating neutrophils suppress T cells and thus prevent them from inhibiting tumor formation. Further, we identify neutrophil-mediated activation of latent TGFβ to be a mechanism of T-cell suppression in mouse adenomas. Finally, we found that neutrophil infiltration is a conserved feature of human CRCs and that concomitant neutrophil infiltration plus TGFβ activation is indicative of T-cell suppression in human tumors.

# Results

### T-cell depletion enhances mouse colon adenoma formation

A variety of mouse intestinal tumor models with distinct properties have been used in CRC research in recent decades (Jackstadt & Sansom, 2016). The majority of these models are based on loss of function of the Apc gene or similar genetic alterations leading to hyperactivation of the Wnt signaling pathway (Jackstadt & Sansom, 2016). Wnt hyperactivation is found in more than 90% of human CRCs (Cancer Genome Atlas, 2012) and is sufficient to induce intestinal adenoma formation in mice (Shibata *et al*, 1997). Most recent studies on mouse tumor formation use Cre recombinase-based mouse models that induce tumor formation predominantly in the small intestine (Jackstadt & Sansom, 2016). Because tumor formation in humans is largely confined to the colon and not the small intestine, we opted to use *Apc*$^{fl/fl-Cdx2CreERT2}$ mice in which Cre activation induces adenoma formation specifically in the colon (Feng *et al*, 2013). Injecting these mice with a low dose of Tamoxifen to activate Cre-recombinase led to the formation of aberrant crypt foci within 3 weeks and the formation of macroscopic adenomatous tumors within 6–14 weeks. Activation of T cells counteracts disease progression in microsatellite instable CRCs (Le *et al*, 2017) and in murine advanced CRC transplantation models (Lau *et al*, 2017), but the role of T cells in colon adenomas has not been investigated so far. To test whether tumor-infiltrating T cells affect colon adenoma formation in *Apc*$^{fl/fl-Cdx2CreERT2}$ mice, we continuously injected them with anti-CD4 and anti-CD8 neutralizing antibodies during and after tumor initiation (Fig 1A). This regimen depleted peripheral T cells and diminished tumor infiltration by about 60% (Fig 1B and Appendix Fig S1A and B). Despite the incomplete depletion of T cells within colon tumors, we observed an increased total tumor volume as a result of increased tumor numbers and a tendency to increased tumor size (Fig 1C). Within the first week of tumor initiation, T-cell depletion had no effect on the number of

cells with increased nuclear and cytoplasmic β-catenin staining (Appendix Fig S1C and D), suggesting that loss of T cells has no effect on the transformation of tumor initiating cells by recombinase-mediated *Apc* gene knockout (Barker *et al*, 2009). T-cell depletion had no effect on the adenomatous phenotype of tumors (Appendix Fig S2A and B). Taken together, this indicates that T cells already counteract tumor formation in the colon of mice at the early adenoma stage, and T-cell depletion accordingly promotes tumor development. To test whether inhibition of tumor formation is mediated via a cytotoxic T-cell response, we injected mice with only anti-CD8 neutralizing antibodies during and after tumor initiation. In contrast to co-depletion of CD4$^+$ and CD8$^+$ T cells, depletion of CD8$^+$ T cells alone had no effect on tumor number or total tumor burden (Appendix Fig S2C and D). This suggests that either CD4$^+$ T cells alone or CD4$^+$ T cells in concert with CD8$^+$ T cells mediate suppression of mouse colon adenoma formation.

### Low-inflammatory T-cell infiltration and high myeloid cell infiltration characterize progression of colon adenomas

Having discovered that T cells counteracted initiation of mouse adenomas, we wanted to investigate whether tumor T cells persisted during adenoma progression. For that purpose, we analyzed the distribution of T cells in tissues from healthy colon and colon tumors by FACS analysis and immunohistochemistry over time. T-cell infiltration was lower in established adenomas than in healthy colon (Fig 2A and B, Appendix Fig S3 and A), suggesting that T cell-exclusion mechanisms develop during adenoma progression. To investigate whether the low tumor T-cell infiltration was based on general exclusion of immune cells or on mechanisms inhibiting T cells specifically, we compared the immune cell contexture of tumors and healthy colon. Infiltration of total CD45$^+$ immune cells between normal colon and tumor tissue was comparable (Appendix Fig S4B–D), contraindicating a general immune cell exclusion from tumors. In contrast to T cells, tumor-infiltrating myeloid cells, including neutrophils, monocytes, and macrophages, were increased compared to normal tissue (Figs 2C–F and EV1A, Appendix Fig S4E). T-cell infiltration in tumor inversely correlated with neutrophil and monocyte infiltration (Fig 2G). This was not the case for neither total macrophages (Fig EV1B), nor macrophages expressing CD206 (Fig EV1C), a marker specific for T cell-suppressive macrophages (Mantovani *et al*, 2017). Investigating inflammatory and immune-suppressive T-cell subsets, we found that both CD8$^+$ and total CD4$^+$ T cells, but not CD4$^+$ Foxp3$^+$ regulatory T cells (Tregs) were decreased in tumors compared to healthy colon (Fig 2H–J). Thus, the fraction of potentially immune-suppressive Tregs among total T cells was increased (Fig 2K). This was not the case for inflammatory interferon-γ (INFγ)$^+$, granzyme B$^+$, or interleukin 17A$^+$ T-cell subsets (Fig EV1D), which means that total numbers of these T-cell subsets are reduced in tumors compared to healthy colon (Fig EV1E). These data indicate a specific reduction of inflammatory T cells in colon tumors and suggest a potential involvement of neutrophils and monocytes in reducing tumor T-cell infiltration.

### Tumor neutrophils suppress T-cell proliferation *in vitro*

Suppression of T cells by neutrophil and monocyte subtypes via various mechanisms have been previously described (Talmadge &

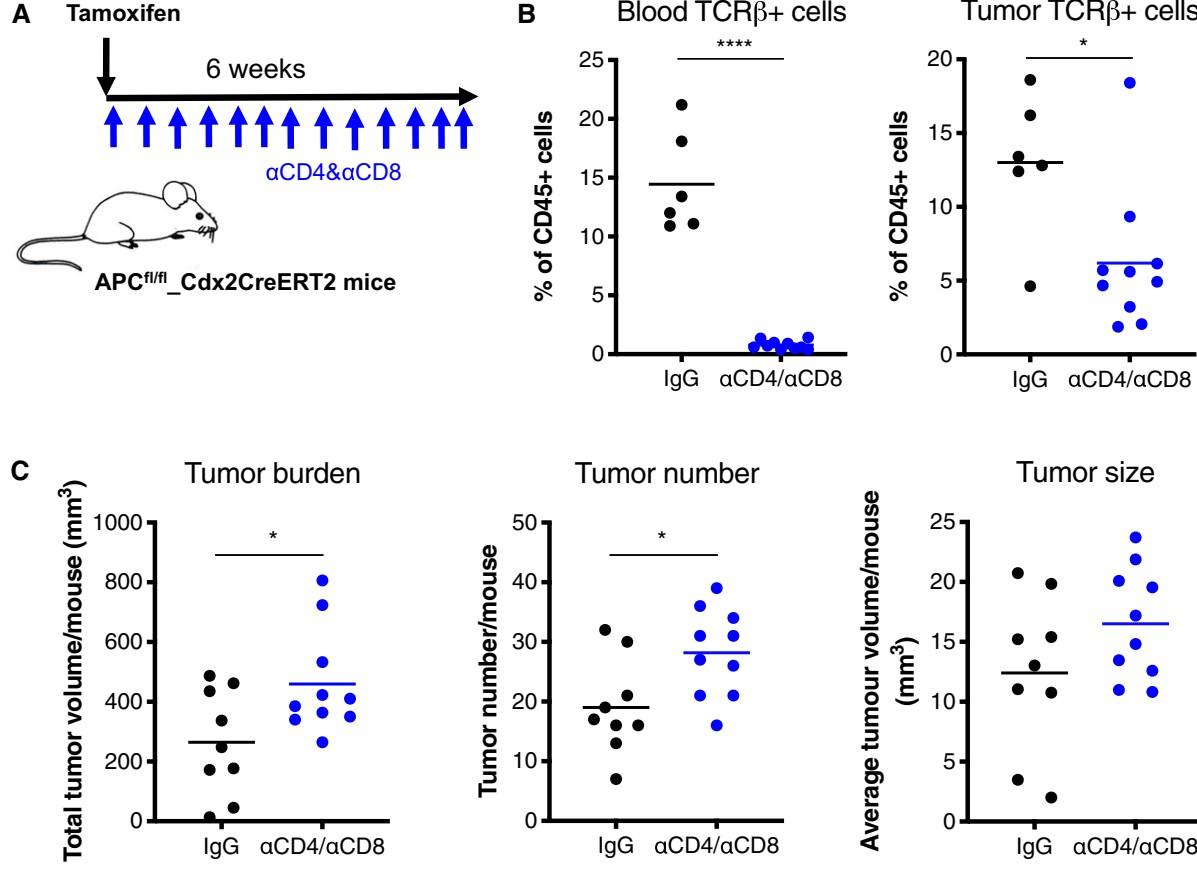

**Figure 1. T-cell depletion leads to increased colon tumor growth in mice.**

A   *Apc^{fl/fl-Cdx2CreERT2}* mice were treated with Tamoxifen and, beginning the day following treatment, injected with either anti-CD4 and anti-CD8 neutralizing antibodies (αCD4/αCD8, blue dots) or IgG control (black dots) twice a week for 6 weeks.

B   FACS analysis of relative TCRβ⁺ T-cell content in blood (left panel) and tumors (right panel) of mice at the end of treatments as indicated in (A). αCD4/αCD8: $n = 10$; IgG: $n = 6$.

C   Tumor burden (left panel), tumor number (middle panel), and average tumor size (right panel) per mouse assessed during colon resection at the end of treatments. αCD4/αCD8: $n = 10$; IgG: $n = 9$.

Data information: In (B–C), statistical analysis was performed by unpaired two-tailed Student's *t*-tests. *$P < 0.05$; ****$P < 0.0001$. Exact *P*-values are provided in Appendix Table S4.

Gabrilovich, 2013), including secretion of immunosuppressive TGFβ and IL10 or enzymatic activity of arginase, nitric oxide synthetase (iNOS), adenosine-converting enzymes (CD39 and CD73), indoleamine-deoxygenase, and cyclooxygenase-2 (PTGS2) (Gabrilovich *et al*, 2012; Bronte *et al*, 2016; Kumar *et al*, 2016; Shaul & Fridlender, 2018). One of the core characteristics of T cell-suppressive myeloid cells is their ability to inhibit T-cell proliferation *in vitro* (Bronte *et al*, 2016). To determine whether any of the tumor-infiltrating myeloid subpopulations in our mouse model exhibit T cell-suppressive activity, we performed co-culture experiments with *in vitro*-activated T cells from healthy mice and tumor-derived myeloid subpopulations (Appendix Figs S3A and S5A–F). Neutrophils and to a lesser extent, macrophages, but not monocytes isolated from colon tumors of mice inhibited proliferation of *in vitro*-activated T cells (Fig 3A), suggesting that tumor-infiltrating neutrophils have high T cell-suppressive potential.

**Neutrophil depletion in tumor-bearing mice reduces tumor burden and activates T cells**

We observed low T-cell infiltration in mouse colon tumors when neutrophils were present. Together with the fact that tumor-derived neutrophils inhibited T-cell proliferation *in vitro*, this indicates that in mouse colon tumors, neutrophils may be involved in T-cell exclusion. To investigate this hypothesis, we treated *Apc^{fl/fl-Cdx2CreERT2}* mice with combined anti-Gr1 antibody and CXCR2 inhibitor at a stage where mice had established tumors with expected high neutrophil and low T-cell infiltration (Fig 3B). This regimen depleted neutrophils, but not monocytes, from blood and tumors of mice (Fig 3C and Appendix Fig S6A and B) and, compellingly, resulted in reduced average tumor size and, consequently, total tumor burden (Fig 3D and Appendix Fig S6C). This correlated with increased tumor infiltration of activated T cells, reduced numbers of Tregs, and a trend to increased total T-cell numbers (Fig 3E–G and

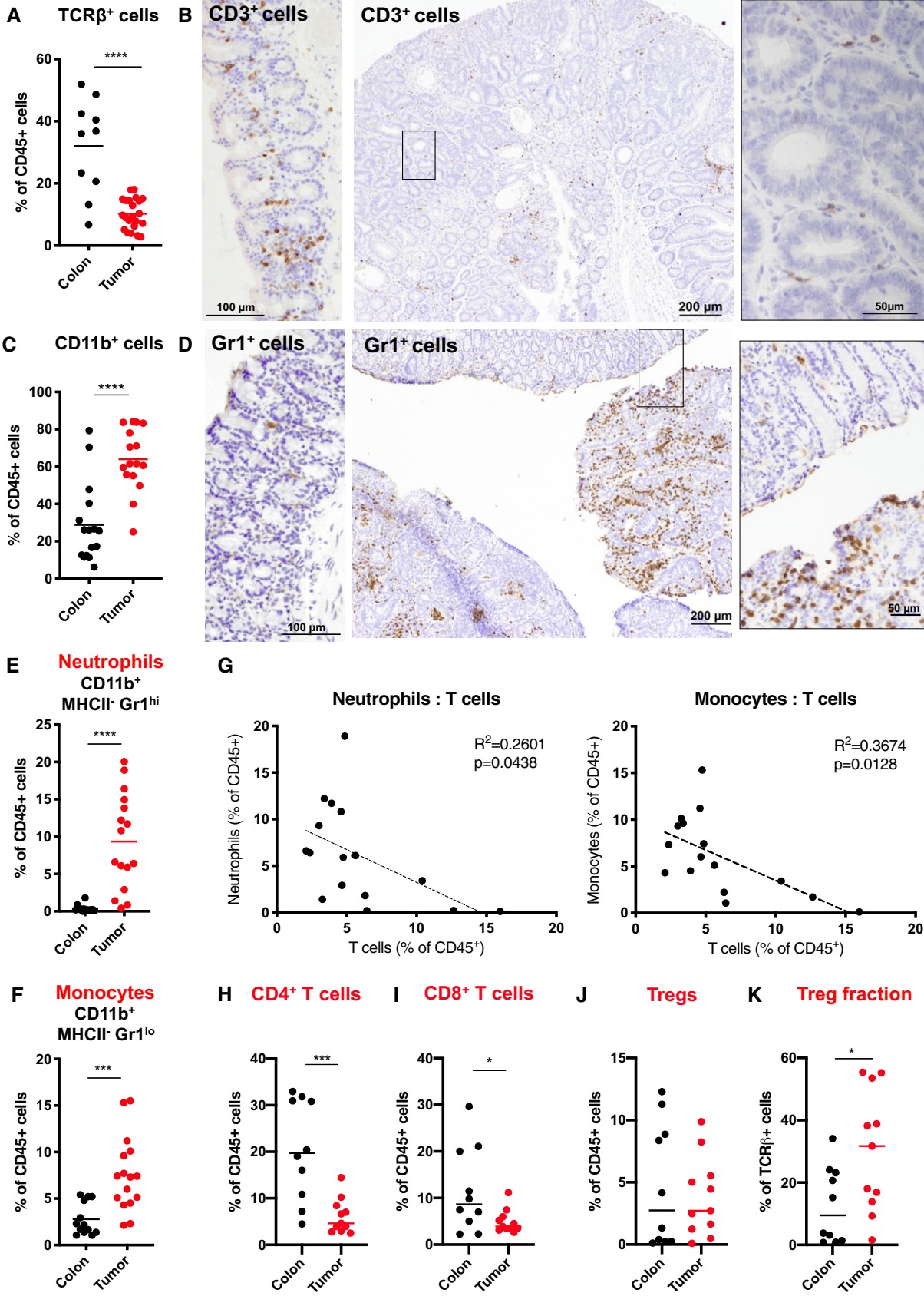

Figure 2.

**Figure 2.  T-cell exclusion in mouse colon tumors correlates with neutrophil and monocyte tumor infiltration.**

Immune cell contexture in healthy colon and colon tumors of *Apc*$^{fl/fl-Cdx2CreERT2}$ mice.

A    Relative TCRβ$^+$ T-cell content in colon ($n$ = 10) and tumor ($n$ = 22).
B    Anti-CD3 immunostaining on normal colon (left panel) and a representative colon tumor section derived from an *Apc*$^{fl/fl-Cdx2CreERT2}$ mouse (right panel: higher magnification of area indicated in middle panel).
C    Relative CD11b$^+$ myeloid cell content in colon ($n$ = 16) and tumors ($n$ = 16).
D    Anti-Gr1 immunostaining on normal colon (left panel) and a representative colon tumor section derived from an *Apc*$^{fl/fl-Cdx2CreERT2}$ mouse (right panel: higher magnification of area indicated in middle panel).
E–G    Relative CD11b$^+$ MHCII$^-$ Gr1$^{hi}$ neutrophil (E) and CD11b$^+$ MHCII$^-$ Gr1$^{lo}$ monocyte (F) content in colon ($n$ = 13) and tumors ($n$ = 16). (G) Correlation of tumor T-cell numbers to sample-matched tumor-derived neutrophil (left panel) and monocyte (right panel) numbers ($n$ = 16).
H–J    Percentage of T-cell subsets among total CD45$^+$ hematopoietic cells in colon ($n$ = 10) and tumor ($n$ = 11). (H) CD4$^+$ TCRβ$^+$ T cells. (I) CD8$^+$ TCRβ$^+$ T cells. (J) Foxp3$^+$ CD4$^+$ TCRβ$^+$ regulatory T cells (Tregs).
K    Percentage of Tregs among total T cells (CD45$^+$ TCRβ$^+$) in normal colon ($n$ = 10) or colon tumors ($n$ = 11).

Data information: In (A), (C), and (E–K), relative content of individual cell types was determined in tissue-derived single cell suspension using cell type-specific antibodies and FACS analysis. Each dot represents pooled tumors from one mouse. (A), (C), (E), (F), and (H–K), statistical analysis performed by unpaired two-tailed Student's *t*-tests. *$P$ < 0.05; ***$P$ < 0.001; ****$P$ < 0.0001. Exact $P$-values are provided in Appendix Table S4. Statistical analysis of (G) was performed by linear regression.
Source data are available online for this figure.

Appendix Fig S6D). In analogy to mice with established colon tumors, treatment of *Apc*$^{fl/fl-Cdx2CreERT2}$ mice with combined anti-Gr1 antibody and CXCR2 inhibitor during and after tumor initiation led to reduced tumor neutrophil infiltration and reduced tumor burden (Fig EV2). When in this experimental setting tumor-infiltrating T cells were co-depleted, neutrophil depletion no longer reduced tumor growth (Fig EV2).

These data strongly suggest that tumor-infiltrating neutrophils impair activation of T cells present in colon tumors and, as such, are part of the tumor growth sustaining tumor microenvironment (TME). In contrast to neutrophil depletion, reducing tumor macrophage infiltration by treating mice with a colony-stimulating factor 1-receptor inhibitor had no effect on tumor burden or tumor T-cell activation (Appendix Fig S6E–G).

**Tumor neutrophils express genes associated with T-cell suppression**

To investigate how neutrophils mediate T-cell suppression, we performed RNA-sequencing on sorted neutrophils and monocytes from tumors and blood of tumor-bearing and healthy mice. Global gene expression was highly altered in tumor-derived cells, indicating that both neutrophils and monocytes undergo substantial transcriptional alterations upon tumor infiltration (Appendix Fig S7A and B). Clear phenotypic differences between tumor associated neutrophils and monocytes and their circulating counterparts have been previously described (Kumar *et al*, 2016; Shaul & Fridlender, 2018). Both tumor-infiltrating neutrophils and monocytes expressed Arginase 2, CD39 and CD73, IL10, iNOS, and Ptgs2 at increased levels compared to circulating cells (Appendix Fig S7C). These genes are all implicated in the secretion of factors known to suppress T cells (Gabrilovich *et al*, 2012; Lanitis *et al*, 2017). Accordingly, transwell co-culture experiment results suggested that the T cell-suppressive properties of neutrophils rely, at least partially, on secreted factors (Appendix Fig S8A). Small molecule inhibitors of Arginase, iNOS, adenosine signaling, Ptgs2, or blocking anti-IL10 antibodies were, however, unable to revert the neutrophil-mediated inhibition of T-cell proliferation in the co-culture assay (Appendix Fig S8B). This is in contrast to a recent study, where inhibition of iNOS partially rescued T-cell suppression mediated by neutrophils isolated from breast tumor-bearing mice (Coffelt *et al*,

2015). Our results, however, suggest that the investigated pathways in the context of mouse colon adenoma are not the principle drivers of the neutrophil-mediated T-cell suppression.

**Neutrophil-secreted MMP9 activates TGFβ-mediated T-cell suppression and tumor promotion**

Since our results and previous studies (Gabrilovich *et al*, 2012; Coffelt *et al*, 2015; Kumar *et al*, 2016) suggest that the most likely mediator of T-cell suppression by neutrophils is a soluble factor, we next aimed to identify these factors *in vitro*. We performed cytokine array analysis of T cells co-cultured with tumor-derived neutrophils or monocytes to identify differentially secreted proteins that may account for the T cell-inhibitory effect of neutrophils. Metalloproteinase 9 (MMP9) and Resistin-like molecule β (RELMβ) were the only two proteins that were specifically identified as being differentially secreted in the T-cell co-culture assay when tumor-derived neutrophils were present (Fig 4A). MMP9 is known to process latent TGFβ into its active form (Yu & Stamenkovic, 2000), and TGFβ was recently shown to be part of the immune-suppressive TME in colon tumors (Mariathasan *et al*, 2018; Tauriello *et al*, 2018). We therefore hypothesized that neutrophils contribute to the inhibitory TME through MMP9 secretion, which subsequently proteolytically cleaves and activates latent TGFβ produced by the TME. To test this hypothesis, we first confirmed the neutrophil-specific expression of MMP9 within tumors by mRNA analysis and co-immunostaining. The major cell type expressing MMP9 mRNA and protein were tumor-infiltrating neutrophils (Fig 4B and C, Appendix Fig S9A and B), while Tgfb1 mRNA expression was mostly restricted to tumor-infiltrating monocytes and Tgfb2, and Tgfb3 mRNA expression was mostly restricted to tumor epithelium and stroma (Fig 4D).

Next, we tested whether inhibition of TGFβ receptor signaling could revert the inhibition of T-cell proliferation in the co-culture assay. Two independent small molecule TGFβ receptor inhibitors (Galunisertib and SB431542) as well as a pan anti-TGFβ antibody rescued T-cell proliferation in the neutrophil co-culture assay (Fig 4E, Appendix Fig S8C). Moreover, and in agreement with the hypothesis, addition of a small molecule MMP inhibitor to the T cell–neutrophil co-culture resulted in the reduction of active TGFβ present in the serum-containing media (Fig 4F), which correlated with increased T-cell proliferation (Fig 4G–H).

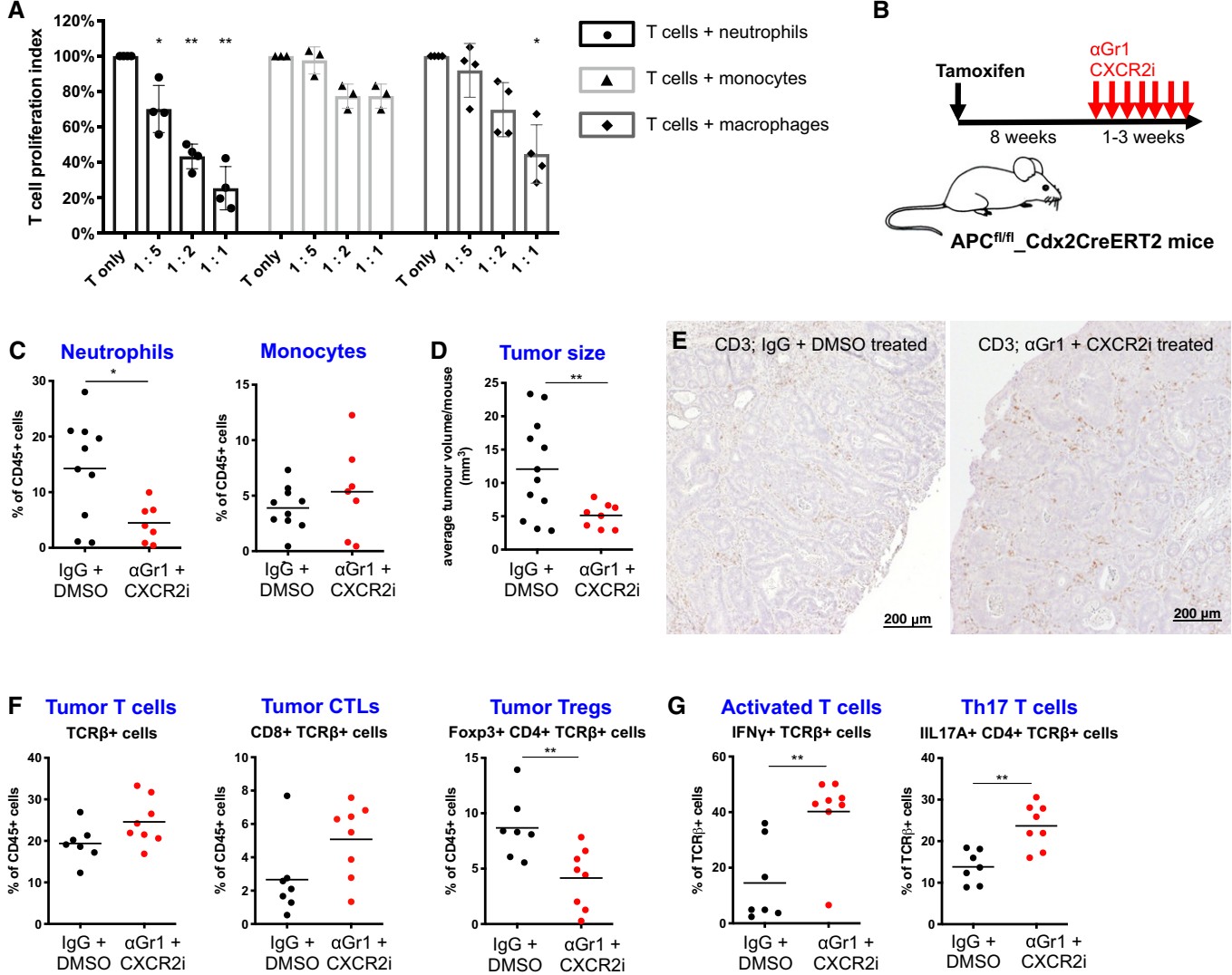

**Figure 3. Tumor neutrophils suppress T cells *in vitro* and *in vivo*.**

A   *In vitro* co-culture of activated T cells with increasing ratios of neutrophils, monocytes, or macrophages. T-cell proliferation index is numbers of proliferated T cells after 3 days of indicated co-culture condition relative to the number of proliferated T cells when cultured alone. CD4$^+$ and CD8$^+$ T cells were derived from lymph nodes of wild-type mice. Neutrophils, monocytes, and macrophages were derived from colon tumors of *Apc$^{fl/fl-Cdx2CreERT2}$* mice. Each dot represents an individual neutrophil (n = 4), monocyte (n = 3), or macrophage (n = 4) sample. Bars represent mean ± SD.

B   Experimental setup. Eight weeks after Tamoxifen treatment of *Apc$^{fl/fl-Cdx2CreERT2}$* mice, animals were treated with anti-Gr1 antibody (αGr1, three times/week) plus CXCR2 inhibitor (CXCR2i, five times/week) or with IgG (three times/week) plus DMSO control (five times/week) for 1–3 weeks.

C   Tumor neutrophil (left panel) and monocyte (right panel) content after αGr1 + CXCR2i (n = 7) or IgG + DMSO (n = 10) treatments.

D   Average tumor volume per mouse at the end of αGr1 + CXCR2i (n = 9) or IgG + DMSO (n = 12) treatments as indicated in (B).

E   Anti-CD3 immunostaining on representative tumor sections derived from mice after indicated treatments.

F   FACS analysis of CD45$^+$ TCRβ$^+$ cells, CD45$^+$ TCRβ$^+$ CD8$^+$ cells, and CD45$^+$ TCRβ$^+$ CD4$^+$ Foxp3$^+$ cells in extracted tumors after αGr1 + CXCR2i (n = 8) or IgG + DMSO (n = 7) treatments.

G   FACS analysis of IFNγ (left panel) and IL17A (right panel) expression in tumor extracted T cells after αGr1 + CXCR2i (n = 7) or IgG + DMSO (n = 7) treatments.

Data information: In (A), statistical analysis was performed by using two-way ANOVA with Dunnett's multiple comparison tests. *$P < 0.05$; **$P < 0.01$. In (C–D) and (F–G), shown are combined data of two independent experiments. Each dot represents pooled tumors from one mouse. Statistical analysis was performed by unpaired two-tailed Student's *t*-tests. *$P < 0.05$; **$P < 0.01$. Exact *P*-values are provided in Appendix Table S4.

Source data are available online for this figure.

To confirm activated TGFβ signaling in mouse colon adenomas, we performed immunostaining for the TGFβ signaling component pSMAD3 and the TGFβ-target gene IGFBP7. A strong staining for pSMAD3 and IGFBP7 could be observed in adenomas compared to benign mucosa (Figs EV3A–D and 5A). pSMAD3 staining within tumors was found in both epithelial and stromal cells (Fig EV3B), while IGFBP7 expression seemed to be restricted to stromal cells (Figs EV3D and 5A). Induction of pSMAD3 and IGFBP7 expression

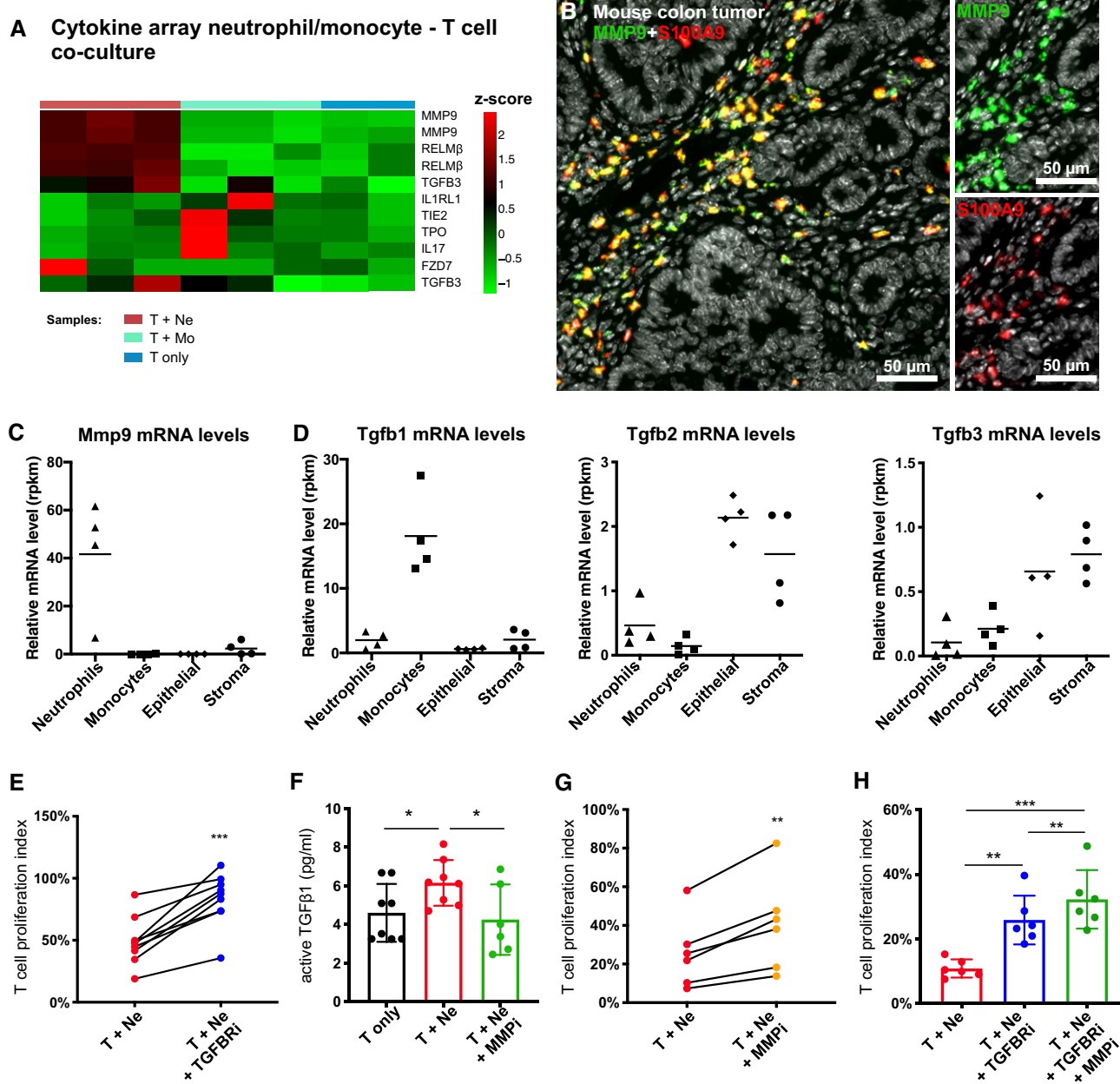

**Figure 4. Tumor neutrophils suppress T cells *in vitro* via MMP9-mediated TGF-beta activation.**

A   Measurement of cytokines present in cell culture supernatants of T cells co-cultured with either tumor-derived neutrophils (T + Ne; *n* = 3) or with tumor-derived monocytes (T + Mo; *n* = 3). Cytokines with a $\log_2$ fold change > 0.6 or < −0.4 were considered as differentially present in supernatants. Shown is a heatmap of scaled cytokines levels (*z*-score) differentially present in T + Ne compared to T + Mo. Adjusted *P*-values were calculated by moderated *t*-test (limma) and Benjamini post-test and were < 0.05 only for MMP9 and RELMβ. Supernatants of T cells alone (T only; *n* = 2) served as internal control. T cells cultured with monocytes did not have differential presence of cytokines compared to T cells alone.

B   Co-immunostaining for MMP9 (green) and S100A9 (red) on a representative tumor section derived from an *Apc*$^{fl/fl\text{-}Cdx2CreERT2}$ mouse. Right panel shows a higher magnification of indicated area of the left panel.

C, D   mRNA expression levels of Mmp9 (C) or Tgfb1, Tgfb2, and Tgfb3 (D) in indicated FACS-sorted cell types isolated from mouse colon tumors (*n* = 4). Data are derived from RNA-sequencing and are displayed as reads per kilo base per million mapped reads (rpkm).

E–H   Neutrophil–T-cell co-cultures treated with vehicle (T + Ne), 10 μm Galunisertib (T + Ne + TGFBRi), 10 μm of the MMP inhibitor GM6001 (T + Ne + MMPi), or with 10 μm Galunisertib plus 10 μm GM6001 (T + Ne + TGFBRi + MMPi). (E, G, and H) T-cell proliferation after 3 days of indicated co-culture conditions. T-cell proliferation index is numbers of proliferated T cells under indicated condition relative to the number of proliferated T cells when cultured alone. (F) Protein levels of free active TGFβ1 in cell culture supernatants of indicated conditions determined by ELISA.

Data information: In (C–H), each dot represents a measurement on an individually derived sample. (E) *n* = 9. (F) T only: *n* = 8; T + Ne: *n* = 8; T + Ne + MMPi: *n* = 6. (G) *n* = 6. (H) *n* = 6. In (E–H), statistical analysis was performed by paired two-tailed Student's *t*-tests. **P* < 0.05; ***P* < 0.01; ****P* < 0.001. Exact *P*-values are provided in Appendix Table S4. (F and H) Bars represent mean ± SD.

Source data are available online for this figure.

was already present in early lesions 3 weeks after tumor induction (Fig EV3C and D) and was accompanied by neutrophil infiltration (Fig EV3C). Depletion of neutrophils in mice reduced protein levels of both pSMAD3 and IGFBP7 within colon tumors (Appendix Fig S9C), suggesting that tumor-infiltrating neutrophils activate TGFβ signaling in colon adenomas throughout their formation. Next, we treated experimental mice *in vivo* with a MMP2/9 inhibitor, which led to reduced numbers of pSMAD3-positive cells, as well as reduced protein levels of remaining pSMAD3-expressing cells within tumors (Fig 5A–C). This suggests that MMP9 produced by tumor-infiltrating neutrophils activates TGFβ signaling within the TME.

Comparable to neutrophil depletion (Figs 3D and EV2D), continuous treatment of mice that develop adenomas with either a TGFβ receptor inhibitor or a MMP2/9 inhibitor directly after adenoma initiation led to reduced average tumor size (Fig 5D and E), suggesting that both MMP9 and TGFβ promote colon adenoma growth.

Taken together, these data indicate that neutrophil-produced MMPs, including MMP9, contribute to the tumor-promoting TME by activating TGFβ, which inhibits tumor-infiltrating effector T-cell activity and is favorable for Tregs (Li & Flavell, 2008) (Fig 5F). Furthermore, our data suggest that these events are initiated very early during tumor formation.

### Combined neutrophil and TGFβ gene signatures associate with absence of a T-cell gene signature in human CRC

Our findings suggest that, in mouse adenomas, neutrophil-mediated activation of latent TGFβ contributes to disease progression by suppressing tumor T cells. The interaction of neutrophils with T cells in human colon tumors has not been investigated until now. Since human CRC develops from adenomas, we hypothesized that the TME of carcinomas may have similar T cell-suppressive mechanisms in place. To investigate whether an interaction of neutrophils with T cells is also relevant in human colon tumors, we first examined a small cohort of 20 archival CRC samples for the presence of neutrophils and T cells by immunostaining, and analyzed publicly available CRC gene expression datasets by cell type-specific gene signatures. S100A9$^+$ and CD66b$^+$ neutrophils were found in all investigated CRC samples. They preferentially localized to the border of the tumor, adjacent to the benign tissue (Fig 6A, Appendix Fig S10). Co-staining for CD8 on the same sections revealed that CD8$^+$ T cells tended to be opposed to neutrophil localization particularly at the tumor border and rarely intermingled (Fig 6B and C, Appendix Fig S10E). This suggests that local presence of neutrophils within CRC tumors might

lead to local exclusion or inhibition of CD8$^+$ T cells. Comparison of T-cell signatures to neutrophil and TGFβ signatures (Appendix Table S1) in two publicly available CRC gene expression datasets revealed that T-cell signatures were lower in tumors with either high neutrophil or high TGFβ signature, but was lowest when both neutrophil and TGFβ signatures were high (Appendix Fig S11A). These data suggest that in human colon tumors where both TGFβ and neutrophils are present, T cells tend to be excluded. We also compared neutrophil, T-cell, and TGFβ signatures to CRC consensus molecular subtypes (CMS1-4), since they delineate tumors partially by the content of their immune infiltration (Guinney *et al*, 2015). Interestingly, the CMS4 tumors that have a high TGFβ signature (Guinney *et al*, 2015; Appendix Fig S11B) and are classified under the immune-suppressed phenotype have the highest neutrophil score among the four CMS tumor subtypes (Fig 6D). When focusing in on CMS4 tumors, the inverse correlation of T-cell signature to neutrophil and TGFβ signatures was even more pronounced (Fig 6E). Thus, patients within the CMS4 subpopulation with a high TGFβ and high neutrophil signature have the lowest T-cell score. Taken together, these correlative data suggest that the presence of neutrophils in CRC subtypes characterized by a high TGFβ signature may significantly contribute to T-cell immune suppression.

### CRC patient-derived neutrophils suppress T cells *in vitro*

To test whether patient-derived neutrophils indeed possess T-cell suppressive potential, we isolated neutrophils and autologous T cells from the blood of treatment naïve CRC patients. Analogous to mouse adenoma neutrophils, CRC patient neutrophils inhibited proliferation of activated T cells when co-cultured *in vitro* (Fig 6F). Moreover, MMP9 expression by neutrophils was consistently detected by immunostaining within human CRC samples (Appendix Fig S12A). MMP inhibitor treatment was surprisingly toxic to human T cells, but treatment of T cell-neutrophil co-culture with TGFBR inhibitor Galunisertib rescued T-cell proliferation at least in part (Fig 6G). These data suggest that the potential to inhibit T cells via TGFβ activation is conserved between neutrophils from mouse colon tumors and CRC patients.

## Discussion

The data presented here strongly suggest that neutrophils suppress anti-tumor T-cell responses through MMP-mediated release of active

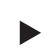

**Figure 5. MMP-induced TGFβ signaling promotes tumor formation.**

A–C  Evaluation of TGFβ signaling activity in mouse colon tumors after four consecutive daily injections of hosts with 40 mg/kg of the MMP2/9-inhibitor SB3-CT (MMPi). (A and B) Co-immunostaining for pSMAD3 (green) and IGFBP7 (magenta) on representative tumor-containing colon sections derived from Apcfl/fl-Cdx2CreERT2 mice treated with vehicle (A) or MMPi (B). (C) Quantification of pSMAD3$^+$ cells (left panel) and pSMAD3 staining intensity per pSMAD3$^+$ cell (right panel) on at least five arbitrarily selected tumor areas per section of MMPi ($n$ = 6)- or vehicle ($n$ = 6)-treated mice.

D, E  Effect of *in vivo* TGFBRi and MMPi treatment of mice for 2 weeks on colon tumor formation. (D) Experimental setup. *Apc$^{fl/fl}$-Cdx2CreERT2* mice were treated with Tamoxifen and, 1 day post-treatment, injected with either MMPi ($n$ = 10) or the TGFBRi SB431542 ($n$ = 10) at 40 mg/kg for 2 weeks with five injections per week. Vehicle-treated mice: $n$ = 11. Average tumor size per mouse was assessed during colon resection 1 week after the end of treatments (E).

F  Model of activation of latent TGFβ stored in extracellular matrix (ECM) by neutrophil-secreted MMP9, leading to suppression of effector T cells and promotion of Tregs.

Data information: In (C) and (E), statistical analysis was performed by unpaired two-tailed Student's *t*-tests. *$P$ < 0.05; **$P$ < 0.01. Exact $P$-values are provided in Appendix Table S4.
Source data are available online for this figure.

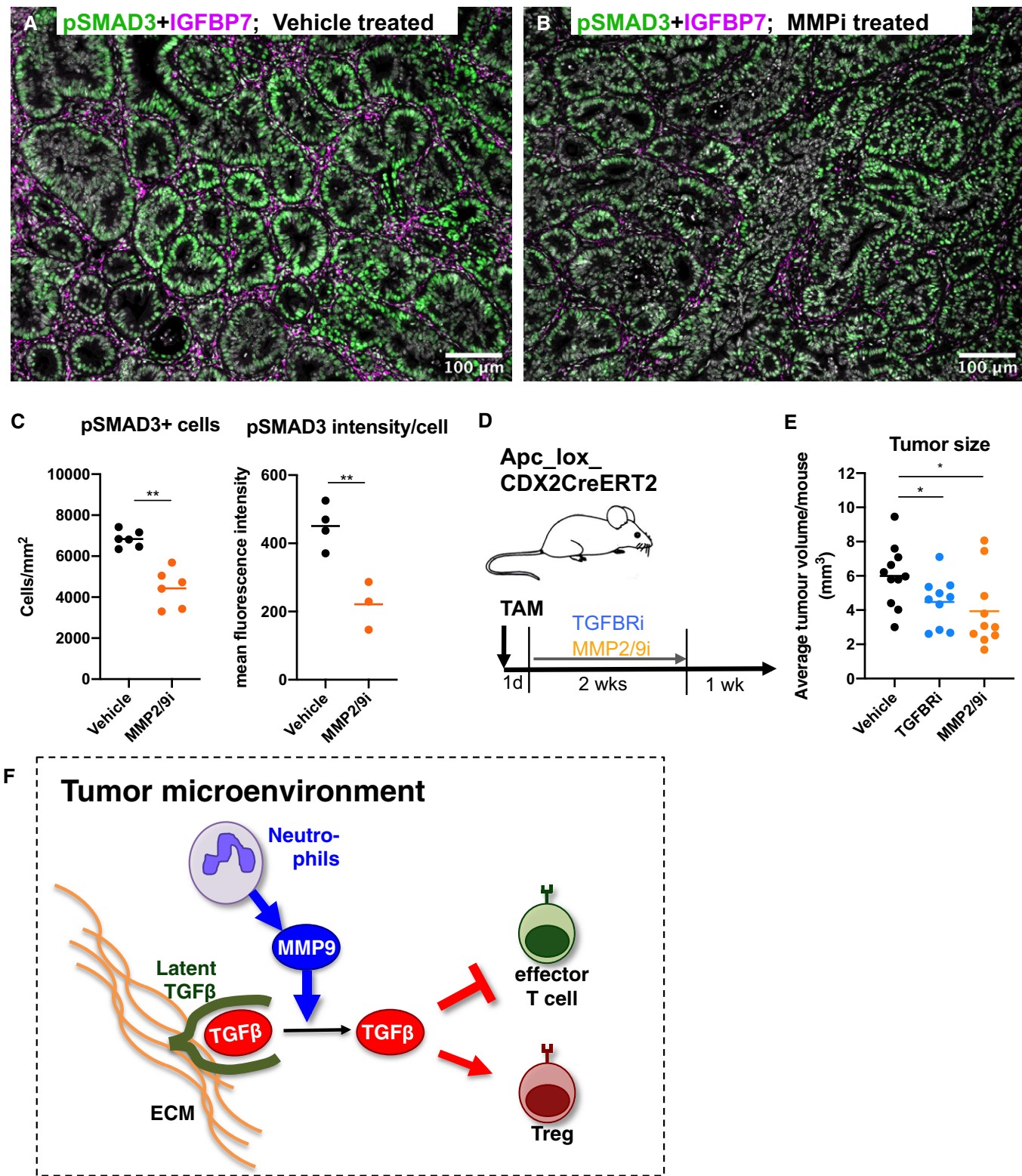

**Figure 5.**

TGFβ within the TME of colon tumors (Fig 5F). T cell-suppressive activity was pronounced in both mouse tumor neutrophils and CRC patient-derived neutrophils. We found neutrophil-mediated TGFβ

activation as a T cell-suppressive mechanism in both species. In large human CRC datasets, tumors with a combined pattern of neutrophil infiltration and TGFβ activation had the lowest T-cell

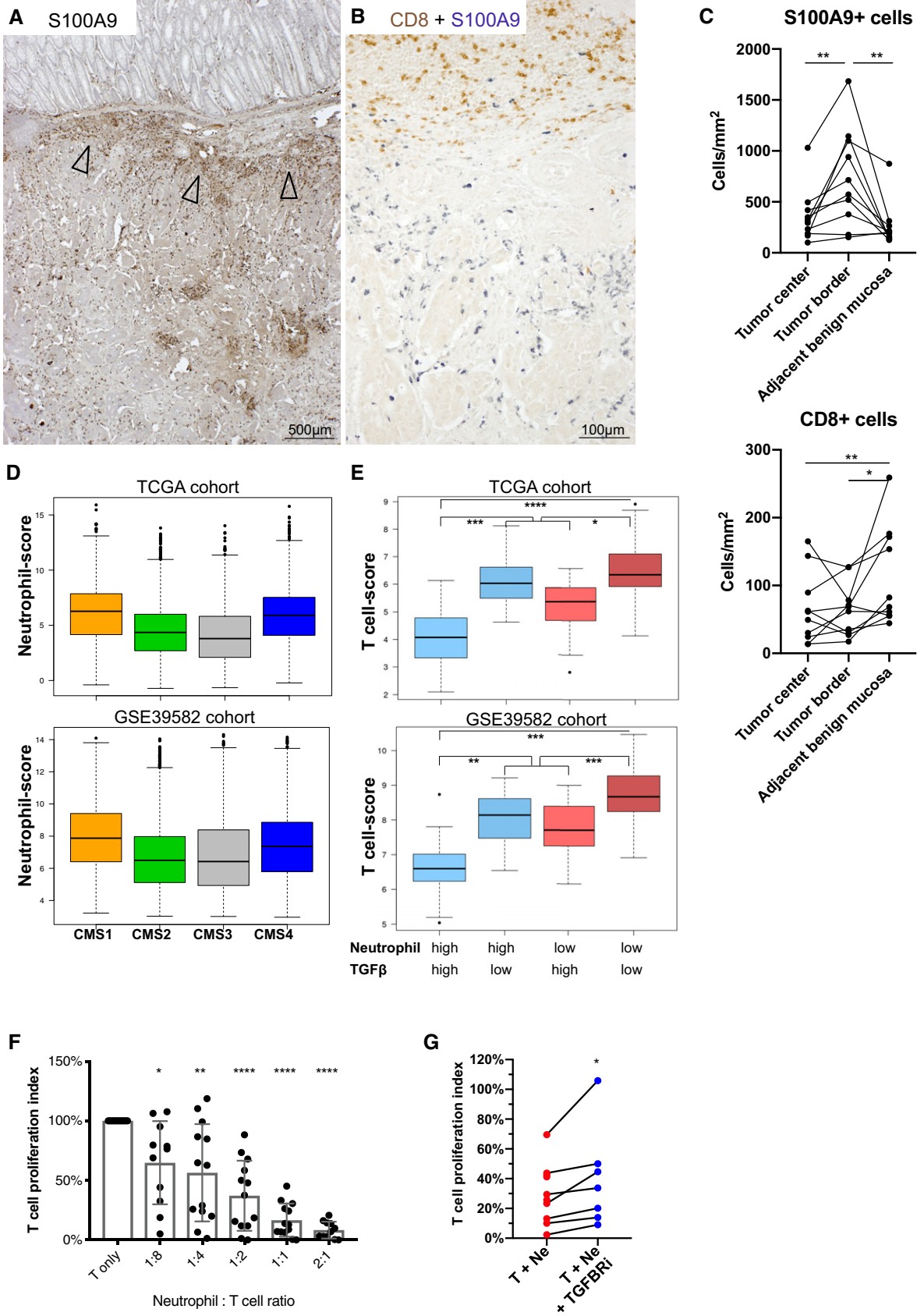

**Figure 6.**

**Figure 6. Neutrophils and TGFβ activity associate with T-cell suppression in human colon cancer.**

A–C Evaluation of S100A9[+] neutrophil and CD8[+] T-cell infiltration in histologic sections of human CRC tumors. (A) Anti-S100A9 immunostaining on a representative human CRC section. Arrowheads indicate neutrophil infiltration at the tumor boarder. (B) Co-immunostaining for CD8 (brown) and S100A9 (gray-blue) in the same CRC sample as shown in (A). (C) Quantification of S100A9[+] cells (upper panel) and CD8[+] T cells in different areas on CRC tumor sections containing tumor border with adjacent benign mucosa as determined by a specialized pathologist: "Tumor center" was tissue within tumor > 200 μm from tumor border, "Tumor border" was tissue within tumor < 100 μm from tumor border, and "Adjacent benign mucosa" was mucosa tissue outside tumor < 100 μm from tumor border. Per section, all identifiable tumor border and adjacent benign areas, as well as three arbitrary tumor center areas, were scored. Sections were derived from 10 individual patients.

D, E Gene expression scores were generated as described in material and methods. Boxes are lower and upper quartiles with median as solid lines; horizontal lines define minimum and maximum; dots define outliers. (D) Neutrophil scores in tumor samples categorized according to CMS subtypes. (E) Comparison of T-cell gene expression scores in CMS4 tumors within two publicly available CRC gene expression datasets. Samples are categorized according to the medians (high = >median; low = <median) of neutrophil and TGFβ gene expression scores. Upper panel: TCGA dataset. Lower panel: GSE39582 dataset.

F Proliferation of activated blood T cells *in vitro* in co-culture with increasing ratios of CRC patient-derived autologous blood neutrophils. CD3[+] T cells and CD66b[+] neutrophils were isolated and cultured individually for each patient. Bar graphs display relative numbers of proliferated T cells after 3 days of co-culture. Each dot per condition represents a neutrophil sample from an individual patient (n = 13). Bars represent mean + SD.

G Proliferation of activated blood T cells *in vitro* in co-culture with autologous blood neutrophils. Cells were derived from both CRC patients (n = 2) and healthy volunteers (n = 5). T cells and neutrophils were cultured at a 1:1 ratio and treated with vehicle (T + Ne) or 10 μm Galunisertib (T + Ne + TGFBRi). Each line represents a neutrophil sample from an individual healthy volunteer.

Data information: In (C) and (G), statistical analysis was performed by paired two-tailed Student's *t*-tests. *$P < 0.05$; **$P < 0.01$. In (D–E), *P*-values were calculated with Mann–Whitney test. *$P < 1 \times 10^{-3}$; **$P < 1 \times 10^{-6}$; ***$P < 1 \times 10^{-8}$; ****$P < 1 \times 10^{-10}$. In (F), statistical analysis was performed by using one-way ANOVA with Dunnett's multiple comparison tests. *$P < 0.05$; **$P < 0.01$; ****$P < 0.001$. Exact *P*-values are provided in Appendix Table S4.

Source data are available online for this figure.

activity signature. We therefore suggest that neutrophil-mediated TGFβ activation is a conserved mechanism of T-cell suppression in human colorectal tumors.

Immune-suppressive as well as context-dependent immune stimulatory properties of neutrophils have been suggested previously for various mouse and human models of inflammation and cancer (Coffelt *et al*, 2016; Shaul & Fridlender, 2018). While the current manuscript was under revision for EMBO Molecular Medicine, two studies were published suggesting a role for neutrophils in the progression of advanced mouse colon cancer (Jackstadt *et al*, 2019; Liao *et al*, 2019), which were in part dependent on the suppression of anti-tumor T cells (Liao *et al*, 2019). A role for neutrophils in immune responses and disease progression has not been described so far for mouse adenomas nor human CRC. Here, we demonstrate that neutrophil depletion in mouse adenomas results in reduced tumor burden and the activation of tumor-infiltrating T cells. Depletion of T cells, in turn, increased tumor burden and abolished beneficial effects of neutrophil depletion. We therefore suggest that neutrophils promote mouse colon tumor progression by suppressing T cells. This is in line with previous studies, suggesting immune-suppressive properties for myeloid cell subsets with a neutrophil-like phenotype (Coffelt *et al*, 2016; Kumar *et al*, 2016; Shaul & Fridlender, 2018). Secreted factors such as TGFβ or IL10, which can directly inhibit T-cell activation, have been suggested as a mechanism of immune suppression mediated by those cells (Coffelt *et al*, 2016; Kumar *et al*, 2016; Shaul & Fridlender, 2018). Here, we propose a novel, indirect mechanism of neutrophil-mediated T-cell suppression that acts via the secretion of MMP9, which in turn leads to activation of TGFβ by proteolytic cleavage.

TGFβ is a major negative regulator of inflammatory responses in general, and of T-cell immune responses in particular (Li & Flavell, 2008; Ortega-Gomez *et al*, 2013). TGFβ protein can be produced by a variety of cell types in healthy tissues or in tumors, but is always secreted as a latent protein (Li & Flavell, 2008). Latent TGFβ can be stored in the extracellular matrix and is commonly activated after proteolytic cleavage by MMPs or other proteinases (Li & Flavell, 2008; Constam, 2014). We show that mouse and human colon tumor neutrophils express high levels of MMP9 and that blocking MMP activity inhibits neutrophil-mediated cleavage of latent TGFβ and T-cell suppression. We thus propose that MMP9 secretion by neutrophils infiltrating colon tumors will automatically lead to TGFβ activation and, consequently, T-cell suppression if latent TGFβ is stored in the TME. Previous studies have suggested activated TGFβ signaling as a T cell-suppressive effect in colorectal cancer and tumor fibroblasts were identified as the main source of TGFβ expression (Calon *et al*, 2015; Tauriello *et al*, 2018). The mechanism by which TGFβ is activated to elicit its T cell-suppressive role in CRC, however, had been unknown.

Immunotherapy using immune checkpoint blockade (ICB) to promote anti-tumor T-cell response is now clinical standard and highly beneficial for cancer patients with various disease origins (Sharma & Allison, 2015; Ribas & Wolchok, 2018). Response to these therapeutics in CRC is limited (Le *et al*, 2017; Overman *et al*, 2017), and an improvement of ICB efficiency is highly desirable (Ribas & Wolchok, 2018). Recently, ICB in combination with TGFBR inhibition was shown to induce an anti-tumor immune response in a metastatic mouse CRC model (Tauriello *et al*, 2018). We suggest that targeting tumor-infiltrating neutrophils could represent another therapeutic option for immunotherapy in CRC. Neutrophils are an essential component of the innate immune defense against pathogens, and thus, neutrophil depletion is not a therapeutic option for cancer patients as they would be at risk of developing severe infections (Metcalf, 2010). Therapies targeting T cell-suppressive cues originating from neutrophils in CRC via CXCR2, TGFBR, or MMP9 inhibitors are attractive options, and the respective compounds are currently under clinical investigation either for CRC (NCT03473925, NCT03470350) or other gastro-intestinal cancer indications (Shah *et al*, 2018). We suggest combination therapy of CXCR2, TGFBR, or MMP9 inhibitors with ICB could broaden immunotherapy responsiveness in CRC patients, especially in those with pervasive neutrophil infiltration and TGFβ activity.

# Materials and Methods

## Ethics statement

All animal work was conducted in accordance with Swiss national guidelines. All mice were kept in the animal facility under EPFL animal care regulations. They were housed in individual cages at 23°C ± 1°C with a 12-h light/dark cycle. All animals were supplied with food and water *ad libitum*. This study has been reviewed and approved by the Service Veterinaire Cantonal of Etat de Vaud. Collection of fresh human samples was conducted under approval of the Swiss Ethics commission.

## Mice

Apc$^{fl/fl}$ mice (Shibata *et al*, 1997) (a kind gift of Tatiana Petrova, University of Lausanne, Switzerland) were crossed to Cdx2CreERT2 mice (The Jackson Laboratory, Charles River, L'Arbresle, France). CreERT2 recombinase was activated at age of 8–12 weeks by a single i.p. injection of 25 mg/kg body weight Tamoxifen (Sigma) in sunflower oil. For T-cell depletion, mice were injected i.p. with a cocktail of anti-CD8 (clone YTS 169.4, in house production) and anti-CD4 (clone YTS 191.1, in house production) neutralizing antibodies at 10 mg/kg body weight each twice a week for the whole duration of the experiment. For neutrophil depletion, mice were injected i.p. with anti-Gr1 neutralizing antibody (clone RB6-8C5, in house production), three weekly injections, 20 mg/kg body weight per dose and with CXCR2-inhibitor SB265610 (Sigma), 5 injections per week, 2 mg/kg body weight per dose. For MMP inhibition, mice were injected i.p. with the MMP2/9 inhibitor SB-3CT (Selleck Chemicals), five injections per week, 40 mg/kg body weight per dose. For TGFBR inhibition, mice were injected i.p. with the ALK4/5 inhibitor SB431542 (MedChem Express), five injections per week, 40 mg/kg body weight per dose. For CSF1R inhibition, mice were injected by oral gavage with the CSF1R-specific inhibitor BLZ945 (AstaTech) dissolved in 20 mg/ml sulfobutylether-β-cyclodextrane pH 2.5 (a kind gift by CycloLab, Budapest, Hungary), five injections per week, 200 mg per kg body weight per dose. Number and size of colon tumors was determined after euthanasia of mice and dissection of their colon. Size of tumors was calculated as (diameter$^3$)/2. Tumor burden was calculated as the sum of volumes of all tumors per mouse. Immune cell types in blood and mesenteric lymph nodes were determined after euthanasia of animals. Blood was drawn by heart puncture.

## Human samples

Archival CRC patient samples were derived from the Institute of Pathology, University of Lausanne, Switzerland. CRC patient blood samples were derived from the Center for Experimental Therapeutics Biobank, University of Lausanne, Switzerland. Blood was drawn at day of surgery before patients underwent resection of primary or metastatic tumors. Blood from healthy donors was derived from the local Blood Transfusion Center, Lausanne, Switzerland. Informed consent was obtained from all subjects and experiments conformed to the principles set out in the WMA Declaration of Helsinki and the Department of Health and Human Services Belmont Report.

## Cell isolation

Mouse tissue-derived cells were isolated from cell suspension after digestion of minced tissue with 1 mg/ml collagenase IV (GIBCO), 0.5 mg/ml Dispase (GIBCO), and 1 mg/ml DNase I (Applichem). Mouse blood cells were derived from peripheral blood after red blood cell lysis. Human immune cells were derived from density gradient separated blood polymorphonuclear cell and mononuclear cell fractions using Lympholyte-poly solution (Cedarlane), and only autologous, sample-matched cells were used for co-culture. Specific cell populations were enriched by either magnetic or fluorescence-based (FACS) sorting for cell type-specific antibodies (see Appendix Tables S1 and S3 for detailed information). Biotinylated antibodies with MACS anti-biotin magnetic beads and MACS columns (Miltenyi Biotec) were used for magnetic sorting. FACS sorting was done on FACS-Aria II or FACS-ARIA Fusion and FACS analysis on LSR Fortessa machines (all BD Biosciences).

## Co-culture assays

Mouse or human T cells were labeled with CellTrace Violet (Thermo Fisher Scientific), plated in 96-well plates, and activated with mouse- or human-specific CD3/CD28-Dynabeads (Thermo Fisher Scientific). Mouse T-cell medium consisted of RPMI1640 + GlutaMAX (GIBCO), 10% fetal bovine serum (GIBCO), 10 μg/ml Gentamycin (GIBCO), 10 mM HEPES (BioConcept), 1 mM Sodium Pyruvate (GIBCO), 1× MEM-NEAA (GIBCO), and 50 μM beta-mercaptoethanol (GIBCO). Human T-cell medium consisted of RPMI1640 + GlutaMAX, 8% human serum (Biowest, France), and 10 μg/ml Gentamycin. Where specified, small molecule inhibitors (see below), neutralizing antibodies (Appendix Table S3), and myeloid cells (neutrophils, monocytes, or macrophages) were added within 1 h after plating of T cells. Small molecule inhibitors were Acetylsalicylic acid (Sigma), Adenosine receptor 2A inhibitor (SCH58261; Cayman Chemicals), Galunisertib (LY2157299; Cayman Chemicals), L-NMMA (Sigma), NorNOHA (Merck-Millipore), MMP inhibitor (GM6001; Merck-Millipore), and SB431542 (Cayman Chemicals). Cells were harvested after 3 days of co-culture and stained with anti-TCRβ or anti-CD3-specific antibodies (Appendix Table S3) and analyzed on a LSR Fortessa FACS machine. T cells with CellTrace violet intensity below freshly stained controls were scored as proliferated T cells.

## Gene expression and pathway analysis of mouse samples

Total RNA was prepared from $2 \times 10^5$ to $5 \times 10^5$ cells and collected in quadruplicates using PicoPure RNA isolation kit (Arcturus). RNA-seq libraries were prepared by first generating double-stranded cDNA from 2.2 ng total RNA with the NuGEN Ovation RNA-Seq System V2 (NuGEN Technologies). 100 ng of the resulting double-stranded cDNA was fragmented to 350 bp using Covaris S2 (Covaris). Sequencing libraries were prepared from the fragmented cDNA with the Illumina TruSeq Nano DNA Library Prep Kit (Illumina) according to the protocol supplied by the manufacturer. Cluster generation was performed with the libraries using the Illumina TruSeq SR Cluster Kit v4 reagents and sequenced on the Illumina HiSeq 2500 with TruSeq SBS Kit v4 reagents. Sequencing data were demultiplexed using the bcl2fastq Conversion Software (v. 2.20, Illumina).

RNA-seq data were mapped as described previously (Langmead & Salzberg, 2012) (mouse genome mm10) and analyzed using the mapping and the RNA-seq modules of HTSstation (David *et al*, 2014). Counts preprocessing and differential analysis were performed with the R packages limma (Smyth, 2004) and voom (Law *et al*, 2014). Genes with an adjusted *P*-value (Reiner *et al*, 2003) lower than 0.05 and an absolute log$_2$ Fold-change > 1.5 were called differentially expressed. Data is publicly available in GEO database (GSE126874).

Gene Set Enrichment Analysis (Subramanian *et al*, 2005) in tumor versus circulating neutrophil and monocytes was performed using gene sets in the Hallmark and KEGG collections of MSigDB v5.0.

### Gene expression analysis in human CRC and adenoma datasets

Datasets: Gene expression and clinical data from human CRC samples in TCGA (RNA-sequencing, $n = 603$) and French (GSE39582, Affymetrix HG133plus2, $n = 566$) cohorts were retrieved from the CRCSC website (Colorectal Cancer Subtyping Consortium, www.synapse.org) (Guinney *et al*, 2015). Samples with identified Consensus Molecular Subtype CMS1, 2, 3, and 4, or only CMS4 (reported as having prominent transforming growth factor β activation) were kept for downstream analysis (TCGA, total cohort $n = 459$, CMS4 subtype $n = 119$; French, total $n = 466$, CMS4 $n = 92$). Gene expression data from human matched colon adenoma and normal colon mucosa were derived from the Gene Expression Omnibus website (www.ncbi.nlm.nih.gov/geo).

### *Defining neutrophil, T cells, and TGFβ scores*

The average log2 expression level of known markers for neutrophils (CXCR1, CXCR2, S100A8, S100A9, CSFR3, MMP9, MMP25, FCGR3B), T cells (CD8A, CD3D, CD3E, CD3G, IRF1, GZMB, IL27, GNLY, PRF1, CCL5, STAT1, IL12RB1, CD28, CCR5, IL12RB2, CD38, CXCR6, TBX21) and TGF-beta activity (DCN, COL1A1, SPARC, ACTA2) were used as prototype for neutrophils, T cells and TGFβ. To expand this list of prototype genes in CRC datasets, multiple linear models were fitted (limma) for each gene in the expression matrix and neutrophils, T cells, and TGF-beta prototype were used simultaneously as explanatory variables. Co-expressing genes with adjusted *P*-values < 0.5 and log fold change > 1 were selected as signatures for neutrophils, T cells, and TGF-beta (with a maximum of 50 genes per signature, Appendix Table S2). To analyze adenoma datasets, prototype signatures were used and were not expanded due to small sample size. Scores for neutrophils, T cells, and TGF-beta activity were computed as the average expression levels of genes in the three signatures.

### *Categories of samples*

Gene expression matrix was normalized for immune content using levels of PTPRC, and then, samples were divided into groups on the basis of neutrophil and TGF-beta scores. First, samples are divided into two groups at the median neutrophil score. Neutrophil-high and neutrophil-low groups are subdivided into two groups on the basis of median TGF-beta score. T-cell scores were plotted in the four groups obtained and *P*-value was computed (Mann–Whitney test).

### Cytokine arrays

Supernatants of mouse activated T cells co-cultured with neutrophils or monocytes, as described above, were collected after 1 day of culture. L-308 mouse antibody arrays (Raybiotech) were used to measure cytokines in supernatants according to instructions by the manufacturer. Array fluorescence signals were analyzed using a G2505C microarray scanner (Agilent Technologies) and ImageJ open source image processing software (https://imagej.net).

### ELISA

Supernatants of mouse activated T cells co-cultured with neutrophils or monocytes, as described above, were collected after 1 day of culture. TGFβ1 protein levels were measured in undiluted supernatant samples using Legend Max Free Active TGFβ1 ELISA kit (BioLegend) according to instructions by manufacturer.

### Immunohistochemistry and immunofluorescence

Paraffin-embedded and fresh frozen tissues were used for histological analysis. For paraffin-embedded tissue, tissues were fixed in 4% PFA overnight at 4°C. Fixed tissue was then processed using standard protocols and embedded in paraffin wax. Four-micrometer-thick sections were cut using a Thermo Scientific Microm HM325 microtome. Sectioned tissue was dried at 37°C overnight and then stored at 4°C. Paraffin-embedded tissue sections were dewaxed and rehydrated. Endogenous peroxidase was blocked in PBS/3% H$_2$O$_2$ for 15 min. Antigen retrieval was then performed with either low pH buffer (citrate pH6) for CD3$^-$, S100A9$^-$, and CD66b-specific antibodies, or with high pH buffer (10 mM Tris, 1mMEDTA, pH9) for CD8$^-$, MMP9$^-$, pSMAD3$^-$, and IGFBP7-specific antibody (see Appendix Table S3 for antibody information).

For fresh-frozen tissue, tissue embedded in OCT compound (TissueTek) using standard protocols. Eight-micrometer-thick sections were cut using a Leica CM1850 cryostat. Cryosections were stored at −80°C. For immunohistochemistry, cryosections were fixed in ice-cold methanol for 2 min. Sections were then rinsed in PBS and incubated in 1% BSA/PBS for 30 min at room temperature to block nonspecific antibody binding.

For paraffin-embedded and fresh-frozen tissues, sections were then blocked in 1% BSA/PBS for 1 h at room temperature. Primary antibody solutions diluted in 1% BSA/PBS were then added and incubated overnight at 4°C or at room temperature for 60 min (see Appendix Table S3 for antibody information). Secondary antibody conjugated with HRP polymers (ImmPRESS, Vector Laboratories) was then added according to instructions by manufacturer. Antibody binding was then revealed either by adding DAB (Sigma) or Vector SG (Vector Laboratories) chromogens for immunohistochemistry, or by using Tyramide Signal Amplification Kits (Thermo Fisher) for immunofluorescence. Microscopy images were acquired with an Olympus AX70 microscope fitted with an Olympus DP70 camera or an Olympus VS120-L100 slide scanner fitted with a Pike F505 C camera. Fluorescent microscopy images were acquired with a Leica DM5500 microscope fitted with a CCD DFC 3000 camera or an Olympus VS120-L100 slide scanner fitted with an Olympus XM10 camera.

For the quantification of immunohistochemically and immunofluorescently labeled cells on histologic sections, slide

## The paper explained

### Problem

Colorectal cancer is one of the major forms of cancer in adults and the second most common cause of cancer death worldwide. Although advances in patient-screening procedures and adjuvant oncological therapies have improved patient outcome, mortality remains significant. High T-cell infiltration in colorectal cancer correlates with a favorable disease outcome and immunotherapy response. This, however, is only observed in a small subset of colorectal cancer patients. A better mechanistic understanding of factors influencing tumor T-cell responses in colorectal cancer could inspire novel therapeutic approaches to achieve broader immunotherapy responsiveness. In the present study, we investigate T cell-suppressive properties of cells and factors present within mouse colon tumors and analyze presence of their human counterpart in colon cancer patient samples.

### Results

We demonstrate that in mouse colon tumors *in vivo*, infiltrating neutrophils possess a remarkable potential to suppress T cells within the tumor microenvironment. One of the recently discovered mechanisms contributing to suppression of T cell-mediated immunity is increased TGFβ levels in the tumor microenvironment. TGFβ, however, is produced as latent inactive pro-protein. How latent TGFβ becomes activated in the tumor microenvironment of colon cancer is unclear. Our results demonstrate that mouse colon–tumor neutrophils secrete matrix metalloproteinases that proteolytically cleave TGFβ into its active form, an event that seems to be initiated very early during tumor formation and conserved during disease progression. Thus, our results provide new mechanistic insight into how neutrophils contribute to an immune-suppressive tumor microenvironment. Neutrophil infiltration is a common feature in human colorectal cancer and colorectal cancer patient-derived neutrophils are able to suppress T cells via TGFβ. Further, analysis of large public datasets indicates that if tumors present with high neutrophil infiltration plus activation of TGFβ, they have the lowest T-cell activation among all patients.

### Impact

Our data suggest that the interaction of neutrophils with a TGFβ-rich tumor microenvironment may represent a conserved immunosuppressive mechanism in colorectal cancer that might be targetable for immunotherapy.

scanner captured images were analyzed using QuPath open source software (Bankhead *et al*, 2017). To analyze per cell staining intensities on slides, immunofluorescently labeled cells, as well as their mean fluorescent signal per cell area, were determined using the Cell Detection algorithm in QuPath software for individual fluorescence channels.

## Statistical analysis

Where not stated otherwise, statistical analysis was performed with GraphPad Prism Version 8 (GraphPad Software Inc., La Jolla, USA). Respective statistical tests used are stated in main text and figure legends.

## Data availability

Mouse mRNA-sequencing data generated during this study have been deposited in GEO database with the primary accession code GSE126874 (https://www.ncbi.nlm.nih.gov/geo/query/acc.cgi?acc=GSE126874).

Codes used in this study (e.g., R functions for gene expression analysis) are freely available from the corresponding authors upon reasonable request.

**Expanded View** for this article is available online.

## Acknowledgements

This work was in part supported by the Swiss Cancer League, the Swiss National Science Foundation, and the PHRT to F.R. We would like to thank Ute Koch for invaluable support, helpful discussions, and critical insights and would like to acknowledge Christelle Dubey, Pasqualina Magliano, and Marianne Nkosi for technical assistance. We would like to thank Tatiana Petrova for providing the Apcfl/fl mouse and Sylvie Rusakiewicz for help with human T-cell culture and providing reagents. We would like to thank Gisele Ferrand for guidance and advice concerning animal experiments, the EPFL histology, microscopy and flow cytometry core facilities for technical assistance, the Lausanne Genomic Technologies Facility hosted by the University of Lausanne for RNA-sequencing and technical assistance on cytokine array analysis, as well as Marion Leleu for help with RNA-sequencing data processing. We would like to thank Olivier Burri and Romain Guiet for assistance on cytokine array image analysis. We would like to thank Olivier Dormond for providing laboratory infrastructure. PW was supported by the Swiss National Science Foundation grant "SNF Project 310030_163351".

## Author contributions

MG designed, performed, analyzed, and interpreted experiments and wrote the manuscript. NZ performed analysis of mouse and human gene expression datasets, as well as analysis of cytokine arrays. M-OS and CS provided and analyzed human archival material. ADB helped to perform and analyze experiments and to write the manuscript. PW and MD helped with human gene expression dataset analysis. ST provided conceptual guidance. LEK and GC provided patient material. FR supervised the project, designed and interpreted experiments and wrote the manuscript. All authors reviewed the manuscript.

## Conflict of interest

The authors declare that they have no conflict of interest.

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
