## [Review Process File · EMBO Molecular Medicine]

Neutrophils suppress tumor infiltrating T cells in colon cancer via matrix metalloproteinase mediated activation of TGF β

Markus Germann, Nadine Zangger, Marc-Olivier Sauvain, Christine Sempoux, Amber D. Bowler, Pratyaksha Wirapati, Lana E. Kandalajt, Mauro Delorenzi, Sabine Tejpar, George Coukos and Freddy Radtke

Review timeline:

Submission date:	28 March 2019
Editorial Decision:	17 April 2019
Revision received:	2 October 2019
Editorial Decision:	24 October 2019
Revision received:	4 November 2019
Accepted:	7 November 2019

Editor: Lise Roth

Transaction Report:

1st Editorial Decision

17 April 2019

Thank you for the submission of your manuscript to EMBO Molecular Medicine. We have now heard back from the referees whom we asked to evaluate your manuscript.

As you will see from the reports below, the 3 referees acknowledge the potential interest and translational relevance of the findings, however they also have fundamental concerns that should be addressed in a major round of revision of the present manuscript. In particular, the mechanism linking neutrophils, MMP9, TGF β and T cells needs to be convincingly strengthened to fully support the conclusions. In several instances, controls, quantifications and/or clarifications are also needed to bring the paper up to publication level.

Addressing the reviewers' concerns in full will be necessary for further considering the manuscript in our journal. EMBO Molecular Medicine encourages a single round of revision only and therefore, acceptance or rejection of the manuscript will depend on the completeness of your responses included in the next, final version of the manuscript.

Please also contact us as soon as possible if similar work is published elsewhere. If other work is published, we may not be able to extend the revision period beyond three months.

I look forward to receiving your revised manuscript.

***** Reviewer's comments *****

Referee #1 (Remarks for Author):

In their manuscript, Germann and colleagues address the immune-suppressive effect of tumor-infiltrating neutrophils in pre-malignant colon adenoma cells. In their APC-floxed, CDX2-cre mouse model, they observe a higher abundance of infiltrating neutrophils in adenoma, and according to their in vitro co-culture model of T cells and neutrophil cells, neutrophil-secreted MMP9 activates latent TGF-beta to mediate effector T cell exhaustion. As has been shown previously by others, increased levels of activated TGF-beta prevent CD8-CTL infiltration and anti-tumor activity in advanced liver metastatic CRC. The authors provide an appealing new mechanism and shed light on how CRC-activated neutrophils mediate this type of immune-suppression at the interface of tumor-associated fibroblasts (which deposit latent TGF-beta in ECM) and T lymphocytes. Also, the finding that CMS4 tumors show a strong neutrophil signature which inversely correlates with the T cell score supports the clinical significance of the in vitro co-culture data presented in the manuscript. However, as an initial and very surprising novel aspect of the study, the authors set out to provide evidence that the functionality of effector T cells already puts a break on pre-malignant colon adenoma initiation/formation. The suggested mechanism of immune-suppression at this early stage of CRC development stays in contrast to what had been observed by others (as specified below), and the here used experimental approach and the actual in vivo experiments performed do in my opinion not justify this conclusion drawn by the authors. There are several weaknesses which should be addressed carefully:

Major point 1:

Since the here used genetic mouse model mainly develops benign colon adenoma with intrinsic Wnt-pathway activation that lack any invasive features, the presented data stand in contrast to observations made by Tauriello et al. (2018, Nature) who reported that T cells indeed infiltrate adenoma extensively and very similar when compared to normal tissue. T cell exclusion was only detectable at the compound mutant (APC, KRAS, TP53, SMAD4) tumor-invasive margins. And only in this area an increased activation of TGF-beta signaling was detected. To address this issue, the authors should bring evidence that latent TGF-beta in the colon adenoma-associated microenvironment of their mice is indeed activated above normal. IHC staining against phospho-SMADs and TGF-beta target genes should be performed on FFPE sections to better characterize the TME of mouse colonic adenoma.

Major point 2:

The authors observe that blocking both CD4 and CD8 positive T-lymphocytes via systemic application of antibodies impacts on mouse colon adenoma formation in the context of Cre-mediated KO of APC. Surprisingly, the main effect of T-cell blockade is an increased number of adenoma frequency while individual adenoma growth/expansion is not significantly altered by continuous application of anti CD4/CD8 antibodies (Figure 1C, right panel). One would expect that a lack of functional CD8 CTL activity supports individual adenoma growth. Blocking neutrophil cells indeed decreased the average tumor volume dramatically, even at later time points (Figure 3D). Together, the presented data rather suggest that neutrophil cells indeed support colon adenoma growth but this might be rather independent of the effector T cell status. The authors should perform additional experiments to support their hypothesis: E.g. blockade of CD4/CD8 or even CD8 lymphocytes alone via antibody application should functionally rescue the observed effect of a lower neutrophil content on in vivo adenoma growth (as depicted in Figure 3C,D).

Major point 3:

Related to point 3: since immune-cells, including T cells, are involved in the integrity of the epithelial barrier function (e.g. reviewed in Konjar S 2017 Front Immunology), the experimental approach chosen by the authors to block both intr CD4 and CD8 cells simultaneously and prior/concomitant with tamoxifen application bears a reasonable risk to produce artifacts. E.g. disturbance of the epithelial barrier might lead to elevated epithelial permeability, an altered bacterial infiltration, and potentially a better accessibility of the colon stem cell niche towards the low dose of tamoxifen applied. This in turn might be an alternative explanation why anti-CD4/CD8 treated animals developed a higher number of adenoma while adenoma growth itself is counter-intuitively not effected (Figure 1C, right panel). The authors should carefully address this technical issue in order to justify the here suggested mechanism of immune-cell function in benign colonic adenoma.

Major point 4:

The authors provide strong but exclusive *ex vivo* evidence via a co-culture system that neutrophil-derived MMP9 released by tumor-associated neutrophils activates latent TGF-beta and hence blocks proliferation and cytotoxic activity of T-cells. Neutrophil-derived MMP9 in liver metastatic CRC has been already described recently by Ogawa and colleagues (Clinical Cancer Research, 2019). Importantly, Germann et al. provide important new and mechanistic data on exactly how this might contribute to modulation of the T cell immune-response.

To provide sufficient evidence that this mechanism indeed plays a role during early pre-malignant colon adenoma initiation/formation *in vivo*, the authors should treat their CDX2-Cre/APC-floxed animals with TGFbeta inhibitors and MMP9 inhibitors directly after adenoma initiation. If their hypothesis is true, then the *de novo* initiated adenoma should get engaged more efficiently by now infiltrating and activated CD8-CTLs, and this should result in a reduced tumor growth/burden.

Minor point 1:

Since the authors observe that presumably CD8-positive effector T cells but not regulatory CD4+ cells might be excluded from colon adenoma tissue (shown in Figure 2K), it would be interesting to see if inhibiting CD8-positive cells alone shows a similar effect on overall adenoma burden.

Referee #2 (Remarks for Author):

Summary

The authors examine the importance of T cells in the development of colon adenomas in a mouse model and identify tumor-infiltrating neutrophils as a mediator of T cell proliferation *in vitro* and activation *in vivo*. They also explore the underlying mechanism by implicating MMP9 and TGFbeta signaling as critical for neutrophil-induced reduction of T cell proliferation *in vitro*. They correlate "high" neutrophil and TGFbeta human gene signatures with "low" T cell signatures, further suggesting the inhibitory role of neutrophils on T cells.

Strengths

The role of immune cells in tumor initiation and development is clinically relevant with the rise in immunotherapies. The authors provide nice evidence for the role of T cells in the early development of colon adenomas. It is important to understand why some colon cancers and not others are able to evade the immune system and the outlook of their studies is timely and interesting.

Weaknesses

This was an ambitious study that was logically presented; however, there was not sufficient evidence for many of conclusions that were drawn. There was also a lack of clarity in the design and presentation of many of the experiments, with some perceived lack of focus. In many cases, additional experiments are needed.

Specific Comments

Abstract/Intro

- The authors fail to use necessary hyphens throughout the paper.
- The authors refer to immune checkpoint blockade inhibitors; however, this is incorrect and should be either immune checkpoint blockade OR immune checkpoint inhibitors.

Results

Figure 1

- The authors should include the gating done for flow cytometry (true for all figures)
- In 1C, the y-axis are inconsistently labeled "tumor" or "tumour"
- It is unclear if the authors are referring to adenoma polyps or malignant tumors in these experiments. This should be clarified.
- The authors should address that depleting T cells with CD4 and CD8 antibodies do not deplete all T cells, as there are double negative T cells that would not be targeted. It is unclear if this population could be important.
- Unpaired t tests would be more appropriate.
- Representative images of the tumor burden in these mice would be helpful.
- It is unclear if TCRb refers to TCRbeta.
- IHC staining showing fewer T cells in these mouse tumors would strengthen these results.

Figure 2

- The benefit of 2B and E is unclear as there is no normal colon tissue staining to compare it to.
- The authors show a decrease in T cells in general (those that are TCRb+) in adenomas, but it would be of additional interest to look at specific T cell populations, such as T helper cells or cytotoxic T cells. Do the ratios of inflammatory vs immunosuppressive T cells change?
- The authors look at CD4+ Tregs, but they should examine the CD8+ Treg population as well.

Figure 3

- It is not clear how the proliferation index was determined.
- It is not clear if the T cells used in the in vitro experiments are primary T cells from mice or a cell line. If they are primary T cells, the authors need to describe how they are isolated and whether they are isolated from WT mice or the APCfl/fl mice? Did they isolate a specific sub-type of T cell or a general T cell population (are they CD4+ or CD8+)? The authors should provide evidence of the purity of the isolated population. This is also necessary for the isolation of neutrophils, monocytes, and macrophages.
- It is not clear why the authors included a CXCR2 inhibitor treatment.
- The authors are not clear on what they believe are the primary suppressive effects neutrophils on T cells in adenomas. Is it through reducing proliferation, which is suggested by the in vitro but not in vivo data, recruitment to the tumor, inhibiting activation, or influencing differentiation into different T cell subtypes?
- The IHC image of increased CD3+ cells with neutrophil depletion (is this 3E, it is not labeled) does not match the FACS analysis in 3F that does not show a change in T cell number.
- The authors should include evaluation of the total tumor number and total tumor burden to be consistent with Figure 1.
- It is unlikely that neutrophils are the only cells influencing T cells in adenomas. The influence of macrophages should not be ignored as the authors provide evidence of their increased numbers in adenomas and ability to reduce T cell proliferation in vitro.
- The conclusion of figure 3 that "...neutrophils impair T cells present in colon tumors..." is too strong, as the data presented do not show any direct reduction in T cell impairment with decreased neutrophils.

Figure 4

- The authors do not adequately prove that MMP9 in the culture media is from neutrophils. Furthermore, they do not provide sufficient evidence of neutrophils as a source of MMP9 in vivo. MMP9 can be released from other cells in the TME.
- Proliferation index needs to be defined.
- It is not clear how "active" TGFbeta was measured.
- The conclusion of this section and the model in 4I is not adequately supported as it is not obvious that the main source of MMP9 is from neutrophils. Additionally the MMP inhibitor used is a pan-inhibitor, so no conclusion can be drawn regarding MMP9 specifically. TGFbeta receptor inhibitors are not very specific and could be inhibiting other pathways. Additional experiments are needed to strengthen the conclusions made in this figure.

Figure 5

- Quantification of IHC staining is necessary for any conclusions to be drawn. This may or may not be a representative image of all tumors.
- It is not clear what is considered "high" vs "low" for the signatures.
- Again, the methods for the isolation of primary cells needs to be more clear and validation of successful isolation needs to be included.

Referee #3 (Remarks for Author):

Germann et al. manuscript EMM-2019-10681:

"Neutrophils suppress tumor infiltrating T cells in colon cancer via matrix metalloproteinase mediated activation of TGFβ"

Summary:

In this study, Germann and colleagues have investigated the mechanisms underlying T cell suppression in colorectal cancer (CRC). Using Apcfl/fl-Cdx2CreERT2 mice that develop adenomas in the colon upon tamoxifen induction, the authors show that co-depletion of CD4+ and CD8+ T

cells increases tumor burden. They demonstrate that myeloid cells make up a large proportion of the immune infiltrate of established tumors, and that of these cells, neutrophils possess most potent T cell-suppressive capacities. Mechanistically, the authors demonstrate that neutrophils produce MMP9, which proteolytically activates TGF β . Interference with this signaling reverses neutrophil-mediated T cell suppression *in vitro*. Lastly, the authors show a correlation between high gene expression profiles of neutrophils, TGF β signaling, and low T cell scores in human CRC. Given the low number of CRC patients responding to immune checkpoint blockers, finding mechanisms that counteract anti-tumor immunity are of substantial importance for the treatment of these patients. This work adds interesting insight into potential mechanisms. However, several aspects need to be addressed:

General comments:

- As the authors discuss in their manuscript, the importance of TGF β in the anti-CRC T cell response has been demonstrated (Tauriello et al., 2018; Mariathasan et al., 2018), the proteolytic activation of TGF β by MMP9 has been described (Yu & Stamenkovic, 2000), and T cell inhibition by neutrophils is well-established (Gabilovich et al., 2012). Although this manuscript proposes a mechanistic link between these aspects, this is only shown in *in vitro* experiments. Without an *in vivo* functional validation of the mechanistic link between neutrophils, MMP9, TGF β and T cells, the novelty of this manuscript is limited. For instance, the authors should check MMP9 and active TGF β protein levels in the tumors of neutrophil-depleted mice. In addition, the authors are advised to assess whether TGF β or MMP inhibition increases T cell proliferation and tumor control *in vivo*.
- In Fig. 3, the authors demonstrate that neutrophil depletion leads to an increase in frequencies of IFN γ + T cells. To formally show that the reduction in tumor size (Fig. 3D) is a result of alleviation of neutrophil-mediated T cell suppression, mice should be treated with CXCR2i + anti-Gr1 in combination with (CD4/CD8) T cell depletion. Otherwise, it cannot be ruled out that other neutrophil-mediated mechanisms control tumor size.
- To support some of the claims in this manuscript, it would be essential to quantify the immunohistochemical analyses shown in Fig. 2, 3, and Supplemental Fig. 1. This is of particular interest because as for example Fig. 3 does not show an increase in T cell frequencies after neutrophil depletion, but the IHC in Fig. 3E demonstrates a dramatic increase in CD3+ cells. This, combined with Fig. 3G, would be important data to demonstrate an immunosuppressive function of neutrophils *in vivo*.
- TGF β blockade reverses neutrophil-mediated immunosuppression *in vitro* (Fig. 4E). However, it is not clear what the source of TGF β is in this setting. *In vivo*, it seems like monocytes and stromal cells produce TGF β . But these are not present in the *in vitro* setting.
- In addition, Fig. 4A shows high TGF β 3, it would be of interest to show the source of TGF β 3, similar to Fig. 4D.
- The authors claim that MMP inhibition in the T cell/neutrophil co-culture reduces active TGF β levels. However, the data shown in Fig. 4F do not show a statistically significant difference. Since a trend of decrease is clearly visible, this experiment would require repetition with more samples to justify this claim.

Minor comments:

- It may be more insightful to separate Fig. 2I into two graphs of neutrophils and monocytes.
- Fig. 4H, these data should be shown in a not-normalized way, to make it more transparent what the extent of the reversal of suppression by TGF β and MMP inhibition is in this culture, compared the proliferation index of the 'T cells alone' group. Alternatively, the figure should include T cells alone in the same normalized fashion.
- Supplemental Fig. 4C would be clarified by adding legends in the figure showing which data comes from circulating and which from tumor-infiltrating immune cells.
- The authors should more clearly state in the methods section where the T cells in their *in vitro* experiments come from, and if these are a mix of CD4+ and CD8+ T cells, or only one of the two populations. And in line with this, it should be more clearly stated what the source of the human T cells and neutrophils is (Fig. 5E, F).
- In the results section, page 5, the authors refer to Figure 1A as "(1A)" instead of "(Fig. 1A)".
- Fig. 3E is missing an "E".

Referee #1:

The authors provide an appealing new mechanism and shed light on how CRC-activated neutrophils mediate this type of immune-suppression at the interface of tumor-associated fibroblasts (which deposit latent TGF-beta in ECM) and T lymphocytes. Also, the finding that CMS4 tumors show a strong neutrophil signature which inversely correlates with the T cell score supports the clinical significance of the in vitro co-culture data presented in the manuscript.

We want to thank this reviewer for the very positive and encouraging comments.

However, as an initial and very surprising novel aspect of the study, the authors set out to provide evidence that the functionality of effector T cells already puts a break on pre-malignant colon adenoma initiation/formation. The suggested mechanism of immune-suppression at this early stage of CRC development stays in contrast to what had been observed by others (as specified below), and the here used experimental approach and the actual in vivo experiments performed do in my opinion not justify this conclusion drawn by the authors. There are several weaknesses which should be addressed carefully:

Major point 1:

Since the here used genetic mouse model mainly develops benign colon adenoma with intrinsic Wnt-pathway activation that lack any invasive features, the presented data stand in contrast to observations made by Tauriello et al. (2018, Nature) who reported that T cells indeed infiltrate adenoma extensively and very similar when compared to normal tissue. T cell exclusion was only detectable at the compound mutant (APC, KRAS, TP53, SMAD4) tumor-invasive margins. And only in this area an increased activation of TGF-beta signaling was detected. To address this issue, the authors should bring evidence that latent TGF-beta in the colon adenoma-associated microenvironment of their mice is indeed activated above normal. IHC staining against phospho-SMADs and TGF-beta target genes should be performed on FFPE sections to better characterize the TME of mouse colonic adenoma.

We agree with the suggested experiment and comment. To address this point, we performed IHC as suggested and stained sections of $Apc^{fl/fl-Cdx2CreERT2}$ mouse colon samples for pSMAD3 and the TGFβ target gene IGFBP7. Strong pSMAD3 staining was detectable in the majority of adenoma epithelial and stromal cells compared to normal colon mucosa, (newly added Fig. EV3A-B). Furthermore, to investigate how early strong pSMAD3 staining was detectable we stained sections of $Apc^{fl/fl-Cdx2CreERT2}$ mouse colon samples just 3 weeks after induction of Cre-mediated recombination of the APC gene. At this stage mice have developed aberrant crypt foci, which also stained positive for pSMAD3 (newly added Fig. EV3C). Similarly—and in agreement with the pSmad3 staining—the TGFβ target gene IGFBP7 was expressed in the stroma of early lesions three weeks after initiation of tumorigenesis, but not in adjacent normal mucosa (newly added Fig. EV3D). Taken together these new experimental data provide strong evidence that elevated TGFβ signaling is indeed a very early event in our mouse model. These data are mentioned in the results section on page 10.

We will briefly comment on the differences between the the study by Taurillo et al and our data. Tauriello et al. indeed show that mouse intestinal carcinomas derived from the AKPT Lgr5-Cre-ERT2 mouse model have low CD3+ cell density and high phosphorylated SMAD3 (pSMAD3) staining, indicative of increased TGFβ pathway activity compared to normal mucosa. Their data further indicates that CD3+ cell density in $Apc^{fl/fl-Lgr5CreERT2}$ adenomas is more similar to normal mucosa than to the compound mutant carcinomas. Nevertheless, their data also indicate that CD3+ cell density is in fact lower in adenomas compared to normal mucosa (Figure 1e in (Tauriello et al, 2018)). It must also be noted that Tauriello et al. investigated mainly adenomas in the small intestine, while our study is restricted to colon adenomas, which is where tumors localize in humans most frequently. Differences between the microenvironments in these two tissues could contribute to variable levels of T cell exclusion. Unfortunately, Tauriello et al. did not investigate TGFβ signaling activation in adenomas in their study, but rather carcinomas.

Major point 2:

The authors observe that blocking both CD4 and CD8 positive T-lymphocytes via systemic application of antibodies impacts on mouse colon adenoma formation in the context of Cre-mediated KO of APC. Surprisingly, the main effect of T-cell blockade is an increased number of adenoma frequency while individual adenoma growth/expansion is not significantly altered by continuous application of anti CD4/CD8 antibodies (Figure 1C, right panel). One would expect that a lack of functional CD8 CTL activity supports individual adenoma growth. Blocking neutrophil cells indeed decreased the average tumor volume dramatically, even at later time points (Figure 3D). Together, the presented data rather suggest that neutrophil cells indeed support colon adenoma growth but this might be rather independent of the effector T cell status. The authors should perform additional experiments to support their hypothesis: E.g. blockade of CD4/CD8 or even CD8 lymphocytes alone via antibody application should functionally rescue the observed effect of a lower neutrophil content on in vivo adenoma growth (as depicted in Figure 3C,D).

We agree with the reviewer that it is somewhat surprising that T cell depletion in established colon adenomas did not lead to a significant change in tumor size. Although tumor size was not significantly increased in this setting, there is a clear trend towards larger tumors.

We also agree with the suggestion to investigate whether the colon adenoma promoting effect of neutrophils is indeed T cell dependent and performed the experiment as suggested. Simultaneous depletion of neutrophils and CD4/CD8 T cells indeed rescued the observed effect of a lower neutrophil content on in vivo adenoma tumor burden. We show this data now as Fig. EV2A-C and mention it in the result section on page 8.

Since we cannot with certainty define whether T cells influence growth or initiation of tumor lesions, we toned down our statement in the Results section on page 5 to “Taken together, this indicates that T cells already counteract tumor formation in the colon of mice at the early adenoma stage, and T cell depletion accordingly promotes tumor development.”

Major point 3:

Related to point 3: since immune-cells, including T cells, are involved in the integrity of the epithelial barrier function (e.g. reviewed in Konjar S 2017 Front Immunology), the experimental approach chosen by the authors to block both intr CD4 and CD8 cells simultaneously and prior/concomitant with tamoxifen application bears a reasonable risk to produce artifacts. E.g. disturbance of the epithelial barrier might lead to elevated epithelial permeability, an altered bacterial infiltration, and potentially a better accessibility of the colon stem cell niche towards the low dose of tamoxifen applied. This in turn might be an alternative explanation why anti-CD4/CD8 treated animals developed a higher number of adenoma while adenoma growth itself is counter-intuitively not effected (Figure 1C, right panel). The authors should carefully address this technical issue in order to justify the here suggested mechanism of immune-cell function in benign colonic adenoma.

To address whether T cell depletion may indeed affect indirectly the frequency of tamoxifen induced Cre-mediated recombination of the floxed APC gene, we quantified recombination events based on increased nuclear and cytoplasmic b-catenin staining, which is only observed as a consequence of loss of the APC gene (Barker et al, 2009). We analyzed β -catenin staining in colon epithelial cells five days after initiation of our T cell depletion regimen. For that purpose, cecum and proximal colon were collected, fixed and sectioned. Ten arbitrary areas per mouse colon section were quantified. We did not observe any significant increase in the recombination efficiency of CD4/CD8 depleted animals compared to IgG treated animals. Taken together these experimental data suggest that T cell depletion does not have an effect on tamoxifen-induced recombination events within the stem cell niche. We added this data as new Appendix Fig. S1C and S1D and describe the results of this experiment on page 5 of the manuscript.

Major point 4:

The authors provide strong but exclusive ex vivo evidence via a co-culture system that neutrophil-derived MMP9 released by tumor-associated neutrophils activates latent TGF-beta and hence blocks proliferation and cytotoxic activity of T-cells. Neutrophil-derived MMP9 in liver metastatic CRC has been already described recently by Ogawa and colleagues (Clinical Cancer Research, 2019). Importantly, Germann et al. provide important new and mechanistic data on exactly how this might contribute to modulation of the T cell immune-response.

To provide sufficient evidence that this mechanism indeed plays a role during early pre-malignant colon adenoma initiation/formation *in vivo*, the authors should treat their CDX2-Cre/APC-floxed animals with TGFbeta inhibitors and MMP9 inhibitors directly after adenoma initiation. If their hypothesis is true, then the *de novo* initiated adenoma should get engaged more efficiently by now infiltrating and activated CD8-CTLs, and this should result in a reduced tumor growth/burden.

We thank the reviewer for the very positive comment and the suggested experiment. First, we treated $Apc^{fl/fl-Cdx2CreERT2}$ tumor bearing mice with the MMP2/9-inhibitor SB-3CT (MMP2/9i) for four days to investigate whether this treatment would indeed result in a reduction of TGFb signaling in adenomas. This indeed resulted in a significant decrease in the number and staining intensity of pSMAD3+ cells within tumors (newly added Fig. 5A-C) indicating that the inhibitor lead to reduced TGFb activation. Next, we treated $Apc^{fl/fl-Cdx2CreERT2}$ tumor bearing mice for two weeks (five days on two days off) with either the MMP2/9-inhibitor SB-3CT or TGFBRi (SB431542) and analyzed tumor size one week post inhibitor treatment. Both, TGFBRi or MMP2/9i treatment led to reduced average tumor size in $Apc^{fl/fl-Cdx2CreERT2}$ tumor bearing mice. This suggests that targeting Tgfb signaling *in vivo* either via inhibition of its receptor or indirectly via MMP inhibition counteracts early adenoma development. This new data is now shown as Fig. 5D-E and are mentioned on page 10 and 11.

Minor point 1:

Since the authors observe that presumably CD8-positive effector T cells but not regulatory CD4+ cells might be excluded from colon adenoma tissue (shown in Figure 2K), it would be interesting to see if inhibiting CD8-positive cells alone shows a similar effect on overall adenoma burden. Our original Figure 2 did not discriminate between exclusion of CD4 and CD8 T cells within the colon adenoma tissue of our mice. We apologize for this. Both CD8⁺ and CD4⁺ FoxP3 negative T cells are excluded from adenomas. This data is now shown Fig. 2H and I (please see also comment of reviewer 2 (Figure 2. Bullet point 2)

We conducted the experiment as suggested and depleted either CD8⁺ T cells alone or in combination with CD4⁺ cells. While co-depletion of CD4⁺ and CD8⁺ cells leads to increased tumor number and total tumor burden (Fig. 1C) depletion of CD8⁺ T cells alone had no effect on tumor number or size (newly added Appendix Fig. S2C-D). This suggests that either CD4⁺ T cells alone or CD4⁺ T cells in concert with CD8⁺ T cells are mediating suppression of mouse colon adenoma formation. These results are now also described on page 5

Referee #2

Summary

The authors examine the importance of T cells in the development of colon adenomas in a mouse model and identify tumor-infiltrating neutrophils as a mediator of T cell proliferation *in vitro* and activation *in vivo*. They also explore the underlying mechanism by implicating MMP9 and TGFbeta signaling as critical for neutrophil-induced reduction of T cell proliferation *in vitro*. They correlate "high" neutrophil and TGFbeta human gene signatures with "low" T cell signatures, further suggesting the inhibitory role of neutrophils on T cells.

Strengths

The role of immune cells in tumor initiation and development is clinically relevant with the rise in immunotherapies. The authors provide nice evidence for the role of T cells in the early development of colon adenomas. It is important to understand why some colon cancers and not others are able to evade the immune system and the outlook of their studies is timely and interesting.

Weaknesses

This was an ambitious study that was logically presented; however, there was not sufficient evidence for many of conclusions that were drawn. There was also a lack of clarity in the design and presentation of many of the experiments, with some perceived lack of focus. In many cases, additional experiments are needed.

We want to thank this reviewer for the encouraging remarks and helpful comments and suggestions.

Specific Comments

Abstract/Intro

- The authors fail to use necessary hyphens throughout the paper.

We apologize and have now added necessary hyphens

- The authors refer to immune checkpoint blockade inhibitors; however, this is incorrect and should be either immune checkpoint blockade OR immune checkpoint inhibitors.

Thank you for pointing this out. We changed the corresponding wording to immune checkpoint blockade/ICB

Results

Figure 1

- The authors should include the gating done for flow cytometry (true for all figures)

Representative gating panels for corresponding analysis and FACS sorting of mouse colon tumor infiltrating hematopoietic cells have been added to Appendix Fig. S3.

- In 1C, the y-axis are inconsistently labeled "tumor" or "tumour"

We apologize and have changed the corresponding wording to "tumor"

- It is unclear if the authors are referring to adenoma polyps or malignant tumors in these experiments. This should be clarified.

Tumors in *Apc^{fl/fl-Cdx2CreERT2}* mice are benign colon adenomas. A corresponding explanation has been added where *Apc^{fl/fl-Cdx2CreERT2}* mice are introduced on page 5 of the manuscript.

- The authors should address that depleting T cells with CD4 and CD8 antibodies do not deplete all T cells, as there are double negative T cells that would not be targeted. It is unclear if this population could be important.

To address this issue, we analyzed the content of CD4⁻ CD8⁻ TCRβ⁺ cells in *Apc^{fl/fl-Cdx2CreERT2}* mouse colon tumors using FACS. Tumors contained virtually no CD4⁻ CD8⁻ T cells, suggesting that double negative T cells are unlikely to play a role in *Apc^{fl/fl-Cdx2CreERT2}* mouse colon tumors. A representative flow cytometric analysis is shown as Appendix Fig. S3C and a more quantitative dot plot figure is shown here, below.

- Unpaired t tests would be more appropriate.

We fully agree. This was a misspelling in the corresponding figure legend; we had indeed used unpaired t tests here. The wording of figure legend has been changed accordingly.

- Representative images of the tumor burden in these mice would be helpful.

As requested, representative images of dissected tumor-bearing ceca have been added in Appendix Figure S2A-B.

- It is unclear if TCRb refers to TCRbeta.

TCRb has been changed to TCR β in Figure 1 panels and legend.

- IHC staining showing fewer T cells in these mouse tumors would strengthen these results.

Images of representative CD3 immunohistochemical staining along with CD3⁺ cell quantification has been added to Appendix Fig S1A and S1B.

Figure 2

- The benefit of 2B and E is unclear as there is no normal colon tissue staining to compare it to.

We agree, CD3 and Gr1 immunohistochemistry of healthy mouse colon has been added to Figures 2B and 2E. Figure 2E of the original manuscript is now Fig. 2D.

- The authors show a decrease in T cells in general (those that are TCRb⁺) in adenomas, but it would be of additional interest to look at specific T cell populations, such as T helper cells or cytotoxic T cells. Do the ratios of inflammatory vs immunosuppressive T cells change?

We agree with the reviewer that a more precise description of T cell subsets in normal colon and colon tumor is of interest. Corresponding panels are now displayed in Fig. 2H – 2K, EV1D-E. A description has been added to the Results section on page 6.

Compared to normal colon, colon tumors contained lower numbers of CD4⁺, CD8⁺, IFN γ ⁺ and CD8⁺ GZMB⁺ T cells (now displayed in Fig. 2H – I and EV1D-E) and tended to be lower for IL17A⁺ T cells. In contrast, the relative number of CD4⁺ Foxp3⁺ regulatory T cells (Tregs) within the CD45⁺ cells did not change (now displayed in Fig. 2J), indicating that the fraction of Tregs among total T cells is increased in tumors (now displayed in Fig. 2K). As discussed in the Results section of the manuscript (page 6), these results suggest that there is a specific loss of inflammatory T cells in tumors compared to healthy colon.

- The authors look at CD4⁺ Tregs, but they should examine the CD8⁺ Treg population as well. To address this issue, we assessed the Foxp3⁺ subsets amongst CD4⁺ and CD8⁺ TCR β ⁺ cells in *Apc^{fl/jf1-Cdx2CreERT2}* mouse colon tumors using flow cytometric analysis. As displayed in the figure below, tumors contained abundant Foxp3⁺ CD4⁺ T cells, but virtually no Foxp3⁺ CD8⁺ T cells.

Figure 3

- It is not clear how the proliferation index was determined.

We agree with the reviewer that the T cell co-culture experiments may not have been described with sufficient clarity. The proliferation index was calculated as numbers of proliferated T cells after three days of indicated co-culture condition relative to the number of proliferated T cells when

cultured alone. Numbers of proliferated T cells was determined as described in Methods section page 17 and 18. A short explanation has been added to the corresponding figure legend on page 33 and 34.

- It is not clear if the T cells used in the *in vitro* experiments are primary T cells from mice or a cell line. If they are primary T cells, the authors need to describe how they are isolated and whether they are isolated from WT mice or the APC^{fl/fl} mice? Did they isolate a specific sub-type of T cell or a general T cell population (are they CD4⁺ or CD8⁺)? The authors should provide evidence of the purity of the isolated population. This is also necessary for the isolation of neutrophils, monocytes, and macrophages.

T cells used for these experiments were a mixture of CD4⁺ and CD8⁺ T cells derived from lymph nodes of wild-type mice using anti-CD4/anti-CD8 MACS isolation. Exact cell isolation procedures are explained in Methods section page 17. A short explanation has been added to the corresponding figure legend on page 33 and 34. Representative purity test panels and corresponding quantifications have been added to Appendix Figures S5A-F and are now mentioned in the Results section on page 7. To measure specifically proliferation of T cells within these samples, we restricted the analysis to TCRβ⁺ cells.

- It is not clear why the authors included a CXCR2 inhibitor treatment.

The reasoning to use this combinatorial treatment was based on preliminary experiments showing that combined treatment of mice with anti-Gr1 plus CXCR2-specific inhibitor SB265610 blocked neutrophil tumor infiltration more efficiently than anti-Gr1 treatment alone (now displayed in Appendix Fig. S6A.).

- The authors are not clear on what they believe are the primary suppressive effects neutrophils on T cells in adenomas. Is it through reducing proliferation, which is suggested by the *in vitro* but not *in vivo* data, recruitment to the tumor, inhibiting activation, or influencing differentiation into different T cell subtypes?

As the reviewer addresses, neutrophils do suppress T cell proliferation *in vitro*. On the other hand, neutrophil depletion *in vivo* led to increased numbers of T cells expressing IFNγ or IL17A (Fig. 3G) and to decreased number of Tregs (Fig. 3F), suggesting that neutrophils may also affect T cell differentiation cues. It has to be noted that IFNγ expression and proliferation are also markers of T cell activation and that T cell differentiation into Tregs likely causes suppression of the activation of other T cell clones. We suggest here that the T cell suppressive effect of neutrophils depends on TGFβ, which is a known regulator of both proliferation and differentiation of T cells (Li & Flavell, 2008). It is therefore difficult to clearly discriminate between T cell proliferation, activation and differentiation in our tumor model, since they likely are overlapping events. We thus believe that the primary suppressive effect of neutrophils on T cells in mouse colon adenomas is inhibition of activation in a broad sense. Discussion of these results has been altered accordingly on page 8

- The IHC image of increased CD3⁺ cells with neutrophil depletion (is this 3E, it is not labeled) does not match the FACS analysis in 3F that does not show a change in T cell number.

We agree with the reviewer that the difference of CD3⁺ cells shown in this picture does not match well the FACS data on T cell infiltration. Analysis of CD3⁺ cells on sections from a series of samples derived from neutrophil depleted or control mice confirmed that there is only a trend of increased numbers of T cells between these two groups of mice (now displayed in Appendix Fig. S6D). Accordingly, histology panels in Fig. 3E have now been changed to more representative images.

- The authors should include evaluation of the total tumor number and total tumor burden to be consistent with Figure 1.

Total tumor number and total tumor burden are now displayed in Appendix Fig. 6C. Mentioning of the reduced total tumor burden resulting from neutrophil depletion has been added to the results section on page 7 and 8

- It is unlikely that neutrophils are the only cells influencing T cells in adenomas. The influence of macrophages should not be ignored as the authors provide evidence of their increased numbers in adenomas and ability to reduce T cell proliferation *in vitro*.

We agree and have therefore performed additional *in vivo* experiments.

Tumor infiltrating macrophages have been described to have either tumor preventive or tumor promoting function (Mantovani et al, 2017). Based on molecular markers they are divided into M1 and M2 subsets and it is generally considered that M2 macrophages are a particularly immune suppressive and tumor promoting macrophage subset (Mantovani et al, 2017). Blockade of colony-stimulating factor 1 receptor (CSF1R) has previously been reported to deplete macrophages in transplantable mouse CRC and fibrosarcoma tumors (Ries et al, 2014) or promote M2 to M1 macrophage polarization in a mouse glioblastoma model (Pyonteck et al, 2013). To test a possible influence of macrophages in the context of T cell suppression and tumor progression in *Apcfl/fl-Cdx2CreERT2* mice, we treated tumor-bearing mice with the CSF1R inhibitor BLZ945 for three consecutive weeks. This treatment led to reduced tumor infiltration of both M1 and M2 macrophages (now displayed in Appendix Fig. S6E). At the same time CSF1R inhibitor treated mice had no significantly different number and average size of tumors and there was no effect on the number of CD4⁺, CD8⁺ or IFN γ ⁺ T cells (now displayed in Appendix Fig. S6F-G). Taken together these results do not support a major role of macrophages in progression nor T cell mediated immune surveillance of *Apcfl/fl-Cdx2CreERT2* mouse colon tumors, now mentioned on page 8.

- The conclusion of figure 3 that "...neutrophils impair T cells present in colon tumors..." is too strong, as the data presented do not show any direct reduction in T cell impairment with decreased neutrophils.

We apologize for the overstatement. What our results demonstrate is increased numbers of IFN γ ⁺ T cells when neutrophils are depleted. We therefore changed our wording to: "tumor-infiltrating neutrophils impair activation of T cells present in colon tumors" on page 8.

Figure 4

- The authors do not adequately prove that MMP9 in the culture media is from neutrophils. Furthermore, they do not provide sufficient evidence of neutrophils as a source of MMP9 *in vivo*. MMP9 can be released from other cells in the TME.

We apologize that this point may not have been described clearly enough. We are, however, confident that the initially presented data demonstrated that MMP9 *in vivo* and *in vitro* is derived from neutrophils. The following data support our claim that MMP9 is produced by neutrophils:

1. Concerning MMP9 protein in *in vitro* experiments, we had measured multiple cytokines in T cell culture supernatants when neutrophils or monocytes had been co-cultured (Fig. 4A). Only when neutrophils were present, high levels of MMP9 could be detected in a cytokine array.
2. Concerning the source of MMP9 *in vivo*; we agree with the reviewer that the immunohistochemical co-staining of MMP9 and the neutrophil marker S100A9 in Fig. 4B may not have been sufficiently clear for the reader. We therefore replaced the picture in panel 4B with an immunofluorescent co-staining of MMP9 and S100A9, which demonstrates a nearly complete overlap of these two markers.
3. Using immunohistochemistry, we now further measured MMP9 expression in tumors of mice where neutrophils had been depleted and found a significant reduction in the number of MMP9⁺ cells (now displayed in Appendix Fig. S9B). These results are now also mentioned in the Results section on page 9 and 10.

- Proliferation index needs to be defined.

The proliferation index was calculated as numbers of proliferated T cells after three days of indicated co-culture condition relative to the number of proliferated T cells when cultured alone. Numbers of proliferated T cells was determined as described in Methods section page 17 and 18. A short explanation has been added to the corresponding figure legend on page 34 and 35.

- It is not clear how "active" TGF β was measured.

Active TGF β in cell culture supernatants was determined using an ELISA assay specific for active TGF β 1 described in Methods section page 20. A short explanation has been added to the corresponding figure legend on page 34 and 35.

- The conclusion of this section and the model in 4I is not adequately supported as it is not obvious that the main source of MMP9 is from neutrophils. Additionally the MMP inhibitor used is a pan-inhibitor, so no conclusion can be drawn regarding MMP9 specifically. TGFbeta receptor inhibitors are not very specific and could be inhibiting other pathways. Additional experiments are needed to strengthen the conclusions made in this figure.

As discussed above, we are confident that our results sufficiently prove neutrophils to be the main source of MMP9 in our experimental systems.

We agree with the reviewer that one caveat of our experiments was the use of a pan-MMP inhibitor in our *in vitro* experiments.

We now performed additional *in vivo* experiments using a novel MMP2/9 inhibitor SB-3CT. We were unable to identify a more specific MMP9 inhibitor. Prolonged treatment of tumor bearing mice with SB-3CT led to reduced TGF β signaling activity and reduced tumor formation (now displayed in Fig. 5B-E). These results are now also mentioned in the Results section on page 10 and 11. As mentioned above, it has to be noted that SB-3CT is a specific inhibitor for both MMP9 and MMP2. MMP2 mRNA, however, is expressed by tumor neutrophils at a very low level compared to MMP9 mRNA (panels displaying MMP2 and MMP9 mRNA levels in tumor neutrophils, tumor monocytes, tumor epithelial cells and tumor stromal cells is shown in the figure here, below). The fact that MMP9 is expressed at the protein level by neutrophils (please see cytokine array and immunostaining in Figure 4A and B) support that the major source of neutrophil-derived MMP activity is MMP9. However, as the inhibitor SB-3CT is bi-specific for both MMP2 and MMP9, we agree with the reviewer that we cannot formally exclude the contribution of additional MMPs such as MMP2. We therefore now mention in our manuscript the MMP2/MMP9 specificity of the inhibitor used in our *in vivo* experiment (Fig.5) and toned down our conclusion on page 11 to :
 “Taken together, these data suggest that neutrophil produced MMPs including MMP9 contribute to the tumor-promoting TME by activating TGF β , which inhibits tumor infiltrating effector T cell activity and is favorable for Tregs (Li & Flavell, 2008) (Fig. 5F). Our data further suggest that these events are initiated very early during tumor formation.”

Figure 5

- Quantification of IHC staining is necessary for any conclusions to be drawn. This may or may not be a representative image of all tumors.

We agree. Tumor tissue and adjacent benign mucosa are infiltrated by both CD8⁺ T cells and S100A9⁺ neutrophils as described before (now displayed in Fig. 6A-B and Appendix Fig. S10). We did quantify S100A9 positive neutrophils and CD8⁺ T cells on sections of human CRC specimens using Qupath, now shown as figure 6C. Compared to benign mucosa, matched tumor tissue has a significantly higher neutrophil infiltration and a significantly lower CD8⁺ T cell infiltration at the

juxtaposed tumor border (now displayed in Fig. 6C). In tumor centers neutrophil numbers are not generally increased compared to matched benign mucosa, but show similarly decreased CD8⁺ T cell infiltration as tumor borders (Fig. 6C).

- It is not clear what is considered "high" vs "low" for the signatures.

Individual samples were scored as "high" or "low" on the respective Neutrophil- or TGFβ-scores depending on whether they were above or below the respective score median value across the whole dataset. This method is explained at length on page 19 and 20 of the manuscript and a short explanation has been added to the corresponding figure legend on page 36 and 37.

- Again, the methods for the isolation of primary cells needs to be more clear and validation of successful isolation needs to be included.

T cells used for these experiments were isolated using anti-CD3 magnetic bead isolation and therefore are likely a mixture of CD4⁺ and CD8⁺ T cells. T cells and autologous neutrophils were isolated from different fractions of the same blood sample. Results shown in Fig. 6F (previously Fig. 5E) are experiments using CRC patient blood-derived neutrophils and T cells. Due to relatively low number of available CRC patient-derived samples, experiments shown in Fig. 6G have been conducted using both CRC patient-derived and healthy volunteer blood samples. Explanations of exact cell isolation procedures are explained in the Methods section on page 17 and have been adapted to improve clarity. A short explanation has also been added to the corresponding figure legend on page 36 and 37. Representative purity test panels and corresponding quantification has been added to Appendix Fig. S12B and are now mentioned in the Results section on page 7.

Referee #3

Summary:

In this study, Germann and colleagues have investigated the mechanisms underlying T cell suppression in colorectal cancer (CRC). Using Apcfl/fl-Cdx2CreERT2 mice that develop adenomas in the colon upon tamoxifen induction, the authors show that co-depletion of CD4⁺ and CD8⁺ T cells increases tumor burden. They demonstrate that myeloid cells make up a large proportion of the immune infiltrate of established tumors, and that of these cells, neutrophils possess most potent T cell-suppressive capacities. Mechanistically, the authors demonstrate that neutrophils produce MMP9, which proteolytically activates TGFβ. Interference with this signaling reverses neutrophil-mediated T cell suppression in vitro. Lastly, the authors show a correlation between high gene expression profiles of neutrophils, TGFβ signaling, and low T cell scores in human CRC. Given the low number of CRC patients responding to immune checkpoint blockers, finding mechanisms that counteract anti-tumor immunity are of substantial importance for the treatment of these patients. This work adds interesting insight into potential mechanisms. However, several aspects need to be addressed:

We thank this reviewer for the positive and encouraging comments.

General comments:

- As the authors discuss in their manuscript, the importance of TGFβ in the anti-CRC T cell response has been demonstrated (Tauriello et al., 2018; Mariathasan et al., 2018), the proteolytic activation of TGFβ by MMP9 has been described (Yu & Stamenkovic, 2000), and T cell inhibition by neutrophils is well-established (Gabilovich et al., 2012). Although this manuscript proposes a mechanistic link between these aspects, this is only shown in in vitro experiments. Without an in vivo functional validation of the mechanistic link between neutrophils, MMP9, TGFβ and T cells, the novelty of this manuscript is limited. For instance, the authors should check MMP9 and active TGFβ protein levels in the tumors of neutrophil-depleted mice.

We agree and thank the reviewer for these helpful suggestions. We performed additional in vivo experiments as suggested. Measuring MMP9 levels using immunohistochemistry revealed a significantly reduced number of MMP9⁺ cells in tumors of neutrophil-depleted mice, along with a reduced number in S100A9⁺ cells (now displayed in Appendix Fig. S9A-B). This data demonstrates that neutrophil depletion does lead to an overall lower MMP9 level in tumors.

As to the best of our knowledge there is no antibody for staining specifically the active forms of TGF β , we decided to use the well-established method of measuring TGF β activity in mouse tissue by immunostaining for the TGF β signaling component pSMAD3 and the TGF β -target gene IGFBP7 (Calon et al, 2015; Jackstadt et al, 2019; Tauriello et al, 2018). Please see also our comments to referee 1 point 1. A strong induction of pSMAD3 and IGFBP7 expression could be observed in adenomas compared to benign mucosa (now displayed in Fig. EV3A-D and Fig. 5A). pSMAD3 staining within tumors was found in both epithelial and stromal cells (now displayed in Fig. EV3B), while IGFBP7 expression seemed to be restricted to stromal cells (now displayed in Fig. EV3D and Fig. 5A). Induction of pSMAD3 and IGFBP7 expression was already present in early lesions three weeks after tumor induction (now displayed in Fig. EV3C-D) and was accompanied by neutrophil infiltration (now displayed in Fig. EV3C). Depletion of neutrophils in mice reduced protein levels of both pSMAD3 and IGFBP7 within colon tumors (now displayed in Appendix Fig. S9C), suggesting that tumor infiltrating neutrophils activate TGF β signaling in colon adenomas throughout their development.

In addition, the authors are advised to assess whether TGF β or MMP inhibition increases T cell proliferation and tumor control in vivo.

We thank the reviewer for the suggested experiment. First, we treated $Apc^{fl/fl-Cdx2CreERT2}$ tumor bearing mice with the MMP2/9-inhibitor SB-3CT (MMP2/9i) for four days to investigate whether this treatment would indeed result in a reduction of TGF β signaling in adenomas. This indeed resulted in a significant decrease in the number and staining intensity of pSMAD3+ cells within tumors (newly added Fig. 5A-C) indicating that the inhibitor lead to reduced TGF β activation. Next, we treated $Apc^{fl/fl-Cdx2CreERT2}$ tumor bearing mice for two weeks (five days on two days off) with either the MMP2/9-inhibitor SB-3CT or TGFBRi (SB431542) and analyzed tumor size one week post inhibitor treatment. Both, TGFBRi or MMP2/9i treatment led to reduced average tumor size in $Apc^{fl/fl-Cdx2CreERT2}$ tumor bearing mice. This suggests that targeting Tgfb signaling either via inhibition of its receptor or indirectly via MMP inhibition counteracts early adenoma development. This new data is now shown as Fig. 5D-E and are mentioned on page 10 and 11.

- In Fig. 3, the authors demonstrate that neutrophil depletion leads to an increase in frequencies of IFN γ + T cells. To formally show that the reduction in tumor size (Fig. 3D) is a result of alleviation of neutrophil-mediated T cell suppression, mice should be treated with CXCR2i + anti-Gr1 in combination with (CD4/CD8) T cell depletion. Otherwise, it cannot be ruled out that other neutrophil-mediated mechanisms control tumor size.

Please see also referee 1 point 2

We agree with the suggestion to investigate whether the colon adenoma growth promoting effect of neutrophils is indeed T cell dependent and performed the experiment as suggested. Simultaneous depletion of neutrophils and CD4/CD8 T cells indeed rescued the observed effect of a lower neutrophil content on in vivo adenoma tumor burden. We show this data now as Fig. EV2A-C and mention it in the result section on page 8.

- To support some of the claims in this manuscript, it would be essential to quantify the immunohistochemical analyses shown in Fig. 2, 3, and Supplemental Fig. 1. This is of particular interest because as for example Fig. 3 does not show an increase in T cell frequencies after neutrophil depletion, but the IHC in Fig. 3E demonstrates a dramatic increase in CD3+ cells. This, combined with Fig. 3G, would be important data to demonstrate an immunosuppressive function of neutrophils in vivo.

We agree with the reviewer that the difference of CD3+ cells shown in these pictures does not match well the FACS data on T cell infiltration. Quantification of CD3+ cells on sections from a series of samples derived from neutrophil depleted or control mice confirmed that there is only a trend of increased numbers of T cells between these two groups of mice (now displayed in Appendix Fig. S6D). Accordingly, histology panels in Fig. 3F were changed to more representative images.

As suggested by the reviewer, we have further quantified immunohistochemical staining of CD3, CD45 and Gr1 for which examples were shown in Fig. 2 and Supplemental Fig. 1 in the originally submitted manuscript. Using this method we could demonstrate:

- A significant reduction of CD3+ cells in colon tumors compared to normal colon mucosa (now displayed in Appendix Fig. S4A), thereby confirming our previous measurement of TCR β + cells by FACS analysis (Fig. 2A).
- Only a minor difference in CD45+ cells between tumor and colon mucosa (now displayed in Appendix Fig. S4B-C), which was comparable to our previous measurement of CD45+ cells by FACS analysis (now displayed in Appendix Fig. S4D). This confirms that the low numbers of T cells in tumors is not caused by a general lack of CD45+ hematopoietic cell infiltration.
- A significant increase in Gr1+ cells in tumors compared to colon mucosa (now displayed in Appendix Fig. S4E), thereby confirming our previous measurement of Gr1+ cells by FACS analysis (Fig. 2E-F).

- TGF β blockade reverses neutrophil-mediated immunosuppression *in vitro* (Fig. 4E). However, it is not clear what the source of TGF β is in this setting. *In vivo*, it seems like monocytes and stromal cells produce TGF β . But these are not present in the *in vitro* setting.

We agree that we did not sufficiently address the source TGF β in our *in vitro* experiment. Bovine serum used in culture medium for *in vitro* experiments contains high amounts of latent TGF β , which can be converted to active TGF β *in vitro*. To verify the content of latent TGF β in our serum-containing T cell culture, we measured concentrations of active and total TGF β 1 in cell culture media using LegendMax Total TGF β 1 ELISA kit (Biolegend). T cell culture medium in the absence of neutrophils contained only very low levels of active TGF β 1, but large amounts of total TGF β 1 (now displayed in Appendix Fig. S8D). This indicates that indeed large amounts of latent TGF β is present in the serum used in our T cell culture experiments, which is then activated through neutrophil secreted MMP as shown in our Figure 4F-H.

- In addition, Fig. 4A shows high TGF β 3, it would be of interest to show the source of TGF β 3, similar to Fig. 4D.

In tumors, TGF β 3 mRNA was mainly expressed by stromal and epithelial cells, while expression in neutrophils and monocytes was low. A corresponding graph has been added to Figure 4D and corresponding information has been added to the manuscript (page 10).

Of note, the expression pattern of Tgfb3 mRNA was very similar to Tgfb2 mRNA, for which the graph in Figure 4D has been changed compared to the initial manuscript version. The reason for this was that the initial version of this graph falsely showed Tgfb2 mRNA levels of cells isolated from healthy tissue instead of tumor derived cells. Compared to the initial graph, the updated version shows that Tgfb2 is expressed not only by stromal cells, but also by epithelial cells in tumors. Since expression Tgfb2 mRNA in tumor neutrophils and monocytes is low, the graph in its current form is still consistent with the message given by the initial graph.

- The authors claim that MMP inhibition in the T cell/neutrophil co-culture reduces active TGF β levels. However, the data shown in Fig. 4F do not show a statistically significant difference. Since a trend of decrease is clearly visible, this experiment would require repetition with more samples to justify this claim.

We agree and have repeated these experiments in order to increase the number of experimental replicates. In accordance with our hypothesis, the reduction of active TGF β 1 protein levels in T cell/neutrophil co-culture in presence of MMP inhibitor reached statistical significance. The combined data are now shown in Figure 4F.

Minor comments:

- It may be more insightful to separate Fig. 2I into two graphs of neutrophils and monocytes.

We agree with the reviewer that separating this data into two graphs might be more suitable for the reader. Consequently, this data is now displayed as two panels in Fig. 2G.

- Fig. 4H, these data should be shown in a not-normalized way, to make it more transparent what the extent of the reversal of suppression by TGF β and MMP inhibition is in this culture, compared the proliferation index of the 'T cells alone' group. Alternatively, the figure should include T cells alone in the same normalized fashion.

We have indeed struggled to display this data in an adequate manner in the original submitted manuscript. We agree with the referee's suggestion and have now repeated these experiments including more samples, which reduced standard deviation for each condition. Therefore, this data is now shown in a non-normalized fashion in Fig 4H.

- Supplemental Fig. 4C would be clarified by adding legends in the figure showing which data comes from circulating and which from tumor-infiltrating immune cells.

We apologize for the lack of a legend in the original submitted figure and have now added such legends to what is now displayed as Appendix Fig. S7C

- The authors should more clearly state in the methods section where the T cells in their *in vitro* experiments come from, and if these are a mix of CD4⁺ and CD8⁺ T cells, or only one of the two populations. And in line with this, it should be more clearly stated what the source of the human T cells and neutrophils is (Fig. 5E, F).

Mouse T cells used for *in vitro* experiments was a mixture of CD4⁺ and CD8⁺ T cells derived from lymph nodes of wild-type mice using anti-CD4/anti-CD8 MACS isolation. Exact cell isolation procedures are explained in Methods section page 17. A short explanation has been added to the corresponding figure legend on page 33 and 34. Representative purity test panels and corresponding quantification has been added to Appendix Figures S5A-F and are now mentioned in the Results section on page 7. To measure specifically proliferation of T cells within these samples, we restricted the analysis to TCR β + cells.

Human T cells used for *in vitro* experiments were isolated using anti-CD3 magnetic bead isolation and therefore are likely a mixture of CD4⁺ and CD8⁺ T cells. T cells and autologous neutrophils were isolated from the same blood sample. Results shown in Fig. 6F (previously Fig. 5E) are experiments with CRC patient derived blood cells. Due to the low number of available CRC patient derived samples, experiments shown in Fig. 6G have been conducted using both CRC patient derived and healthy volunteer blood samples. Explanations of exact cell isolation procedures in Methods section page 17 have been adapted to improve clarity. A short explanation has been added to the corresponding figure legend on page 36 - 37. Representative purity test panels and corresponding quantification has been added to Appendix Fig. S12B.

- In the results section, page 5, the authors refer to Figure 1A as "(1A)" instead of "(Fig. 1A)".

We would like to thank the reviewer for this comment and have changed the formulation on page 5 accordingly.

- Fig. 3E is missing an "E".

We thank the reviewer for this comment and have added the labeling in Figure 3E accordingly.

References mentioned in the point by point response

Barker N, Ridgway RA, van Es JH, van de Wetering M, Begthel H, van den Born M, Danenberg E, Clarke AR, Sansom OJ, Clevers H (2009) Crypt stem cells as the cells-of-origin of intestinal cancer. *Nature* **457**: 608-611

Calon A, Lonardo E, Berenguer-Llargo A, Espinet E, Hernando-Momblona X, Iglesias M, Sevillano M, Palomo-Ponce S, Tauriello DV, Byrom D, Cortina C, Morral C, Barcelo C, Tosi S, Riera A, Attolini CS, Rossell D, Sancho E, Batlle E (2015) Stromal gene expression defines poor-prognosis subtypes in colorectal cancer. *Nature genetics* **47**: 320-329

Jackstadt R, van Hooff SR, Leach JD, Cortes-Lavaud X, Lohuis JO, Ridgway RA, Wouters VM, Roper J, Kendall TJ, Roxburgh CS, Horgan PG, Nixon C, Nourse C, Gunzer M, Clark W, Hedley A, Yilmaz OH, Rashid M, Bailey P, Biankin AV, Campbell AD, Adams DJ, Barry ST, Steele CW, Medema JP, Sansom OJ (2019) Epithelial NOTCH Signaling Rewires the Tumor Microenvironment of Colorectal Cancer to Drive Poor-Prognosis Subtypes and Metastasis. *Cancer cell* **36**: 319-336 e317

Li MO, Flavell RA (2008) TGF-beta: a master of all T cell trades. *Cell* **134**: 392-404

Mantovani A, Marchesi F, Malesci A, Laghi L, Allavena P (2017) Tumour-associated macrophages as treatment targets in oncology. *Nature reviews Clinical oncology* **14**: 399-416

Oida T, Weiner HL (2010) Depletion of TGF-beta from fetal bovine serum. *Journal of immunological methods* **362**: 195-198

Pyonteck SM, Akkari L, Schuhmacher AJ, Bowman RL, Sevenich L, Quail DF, Olson OC, Quick ML, Huse JT, Teijeiro V, Setty M, Leslie CS, Oei Y, Pedraza A, Zhang J, Brennan CW, Sutton JC, Holland EC, Daniel D, Joyce JA (2013) CSF-1R inhibition alters macrophage polarization and blocks glioma progression. *Nature medicine* **19**: 1264-1272

Ries CH, Cannarile MA, Hoves S, Benz J, Wartha K, Runza V, Rey-Giraud F, Pradel LP, Feuerhake F, Klaman I, Jones T, Jucknischke U, Scheiblich S, Kaluza K, Gorr IH, Walz A, Abiraj K, Cassier PA, Sica A, Gomez-Roca C, de Visser KE, Italiano A, Le Tourneau C, Delord JP, Levitsky H, Blay JY, Ruttinger D (2014) Targeting tumor-associated macrophages with anti-CSF-1R antibody reveals a strategy for cancer therapy. *Cancer cell* **25**: 846-859

Tauriello DVF, Palomo-Ponce S, Stork D, Berenguer-Llargo A, Badia-Ramentol J, Iglesias M, Sevillano M, Ibiza S, Canellas A, Hernando-Momblona X, Byrom D, Matarin JA, Calon A, Rivas EI, Nebreda AR, Riera A, Attolini CS, Batlle E (2018) TGFbeta drives immune evasion in genetically reconstituted colon cancer metastasis. *Nature* **554**: 538-543

2nd Editorial Decision

24 October 2019

Thank you for the submission of your revised manuscript to EMBO Molecular Medicine. The manuscript was sent back to the three reviewers, and as you will see from the reports below, they are now supportive of publication pending minor revisions. Therefore, we would like you to address the referees' comments as well as the following editorial amendments before acceptance of your manuscript for publication in EMBO Molecular Medicine:

1) Referees' comments:

Please address both referees #1 and #3's comments in writing. At this stage, we'd like you to discuss the referees' points, and if you do have data at hand, we'd be happy for you to include it, but we will not require any additional experiments. However, regarding referee #3's comment on the different treatment schedules, if you do not provide new data, please make sure you carefully explain and discuss the limitations in your manuscript.

I look forward to reading a new revised version of your manuscript as soon as possible.

***** Reviewer's comments *****

Referee #1 (Remarks for Author):

In their revised manuscript, Germann et al. have carefully addressed the concerns raised on the original/initial version. New and critical experiments have been performed and interpreted correctly, technical issues have been clarified, and the conclusions drawn were adapted and are now justified by the data provided.

The role of activated TGF-beta signaling and MMP activity during early adenoma development has now been analyzed in vivo by showing elevated TGF-beta signaling activity in adenomatous mouse tissue (Fig. EV3A-C) and treatment of the mice with chemical inhibitors (Fig. 5A-E). This brings important evidence for the biological relevance of the data derived from the in vitro co-culture experiments. Furthermore, the importance of neutrophil cells for mouse colonic adenoma growth has been strengthened by additional data, such as the blockade of macrophage activity by application of a CSF1-R inhibitor which could not mimic the effect of neutrophil blockade. The data suggest that in the here used mouse model, colon adenoma infiltrating macrophages are presumably not pro-tumorigenic in a non-inflammatory environment and do not influence the T cell status as shown in Appendix Fig. S6E-G. Since most published colon adenoma mouse models were induced via a combined DSS treatment, which might provide macrophages with different traits/activities, these data are especially interesting for a broad readership in the field. The rescue experiment shown in Figure EV2A-C, showing that blockade of CD4/CD8 cells reversed the effect of a lower neutrophil content on adenoma burden, nicely demonstrates the functional interplay between these two immune cell types in the control of early adenomatous tumor development.

Minor point:

It is interesting that CD8+ T cells alone do not seem to mediate the effect on adenoma development. At least their specific blockade did not affect adenoma development (Appendix Fig. S2C,D). CD4+/FoxP3-neg T cells might be equally important or even sufficient for the anti-tumorigenic effect according to the new data provided. Although the authors did unfortunately not show depletion of CD4+ cells alone (I did actually ask only for depletion of CD8+ cells alone), the new data highlight the importance of CD4+ cells, alone or in concert with CD8+ cells, for the control of early adenoma development. However, in Figure 6 the authors correlate higher neutrophil levels in patient tumor tissues compared to adjacent normal tissue only with a lower level of infiltrating CD8+ T cells. It would be very interesting to analyze also the status of CD4+ T cells in this context, since co-blockade of CD4+ cells with CD8+ cells was essential to augment adenoma growth in the mouse colon adenoma model.

Overall, the additional data provided by the authors have significantly improved the quality of the manuscript. In my opinion the revised manuscript version should be considered for publication in EMBO Molecular Medicine.

Referee #2 (Comments on Novelty/Model System for Author):

the authors have adequately addressed the comments of the reviewers and the manuscript is now suitable for publication

Referee #2 (Remarks for Author):

The authors have adequately addressed the comments of the reviewers and made the needed revisions to the manuscript.

Referee #3 (Comments on Novelty/Model System for Author):

This is fundamental research, therefore the immediate medical impact is yet moderate. But only by fundamental research like performed in this ms, we will in the future make medical impact.

Referee #3 (Remarks for Author):

Germann et al. manuscript EMM-2019-10681:
 "Neutrophils suppress tumor infiltrating T cells in colon cancer via matrix metalloproteinase mediated activation of TGFβ"

In the revised manuscript, the authors have added new experimentation to address most of the Reviewer's comments and to strengthen the conclusions.

I have concerns about the treatment schedule of the newly added experiments. For instance, in newly added Figure EV2 the authors have combined neutrophil depletion with T cell depletion. These experimental groups are supposed to link with the experiment shown in Figure 3. However, based on the information in the legend of figure EV2, the treatment schedule is very different from that shown in Fig. 3B. In Fig. EV2, the treatment starts 1 day after tamoxifen induction (thus, during the early adenoma phase). Whereas the important message of Figure 3 is that mice with established tumors are being treated. In the text page 8, the authors raise the suggestion that both experiments are performed under the same conditions, which is incorrect. The experiment shown in Fig EV2 should be performed under the same conditions as the experiment in Fig. 3, so the combined depletion and controls (also the T cell depletion only group) should start 8 weeks after tamoxifen induction.

Same concern is true for the experiment in Figure 5D, E. Why did the authors not follow the same treatment schedule as in Figure 3, so treatment of established tumors? In the early adenomas, T cell suppression is less of an issue, as shown in Figure 1, where T cells do limit adenoma formation. These different treatment regimens reduce the transparency of the paper (e.g. due to different treatment schedule, the reader cannot compare the tumor volume of fig. 5E with that of Fig 1C)

2nd Revision - authors' response

4 November 2019

Referee #1 (Remarks for Author):

In their revised manuscript, Germann et al. have carefully addressed the concerns raised on the original/initial version. New and critical experiments have been performed and interpreted correctly, technical issues have been clarified, and the conclusions drawn were adapted and are now justified by the data provided.

The role of activated TGF-beta signaling and MMP activity during early adenoma development has now been analyzed in vivo by showing elevated TGF-beta signaling activity in adenomatous mouse tissue (Fig.EV3A-C) and treatment of the mice with chemical inhibitors (Fig.5A-E). This brings important evidence for the biological relevance of the data derived from the in vitro co-culture experiments. Furthermore, the importance of neutrophil cells for mouse colonic adenoma growth has been strengthened by additional data, such as the blockade of macrophage activity by application of a CSF1-R inhibitor which could not mimic the effect of neutrophil blockade. The data suggest that in the here used mouse model, colon adenoma infiltrating macrophages are presumably not pro-tumorigenic in a non-inflammatory environment and do not influence the T cell status as shown in Appendix Fig. S6E-G. Since most published colon adenoma mouse models were induced via a combined DSS treatment, which might provide macrophages with different traits/activities, these data are especially interesting for a broad readership in the field.

The rescue experiment shown in Figure EV2A-C, showing that blockade of CD4/CD8 cells reversed the effect of a lower neutrophil content on adenoma burden, nicely demonstrates the functional interplay between these two immune cell types in the control of early adenomatous tumor development.

Minor point:

It is interesting that CD8+ T cells alone do not seem to mediate the effect on adenoma development. At least their specific blockade did not affect adenoma development (Appendix Fig. S2C,D). CD4+/FoxP3-neg T cells might be equally important or even sufficient for the anti-tumorigenic effect according to the new data provided. Although the authors did unfortunately not show depletion of CD4+ cells alone (I did actually ask only for depletion of CD8+ cells alone), the new data highlight the importance of CD4+ cells, alone or in concert with CD8+ cells, for the control of early adenoma development. However, in Figure 6 the authors correlate higher neutrophil levels in patient tumor tissues compared to adjacent normal tissue only with a lower level of infiltrating CD8+ T cells. It would be very interesting to analyze also the status of CD4+ T cells in this context, since co-blockade of CD4+ cells with CD8+ cells was essential to augment adenoma growth in the mouse colon adenoma model.

Overall, the additional data provided by the authors have significantly improved the quality of the

manuscript. In my opinion the revised manuscript version should be considered for publication in EMBO Molecular Medicine.

We thank this referee for their very positive comments and suggestions. The CD8-depletion experiment was done in the context of the revision to this manuscript. Like the referee, we also expected that depletion of CD8 T cells alone would result in increased tumor burden. Thus, we were as surprised as the referee about this finding. We agree that in this context the depletion of CD4⁺/FoxP3^{neg} T cells would have been interesting. However, the results of an antibody-mediated CD4-depletion experiment would have been difficult to interpret because in such an experiment one cannot distinguish between the effects mediated by depletion of helper (CD4⁺ FoxP3⁻) and regulatory (CD4⁺ FoxP3⁺) T cells as both of them would be depleted. Thus, selective depletion of T helper cells is not trivial. Theoretically we could have tried to selectively deplete Tregs using DERE mice, which express a diphtheria toxin receptor under control of the FoxP3 locus. Although interesting, this experiment was just not feasible within a reasonable time frame for the revision to this manuscript. We therefore have not performed any anti CD4-mediated T cell depletion experiments.

Because of the known importance of CD8 T cells within human colorectal cancer specimens (Pages et al, 2009) we did not perform any staining's for CD4 T cell infiltration on human colon cancer specimens and therefore have no data at hand that could be easily incorporated in the current version of the manuscript.

Referee #2 (Comments on Novelty/Model System for Author):

the authors have adequately addressed the comments of the reviewers and the manuscript is now suitable for publication

Referee #2 (Remarks for Author):

The authors have adequately addressed the comments of the reviewers and made the needed revisions to the manuscript.

Referee #3 (Comments on Novelty/Model System for Author):

This is fundamental research, therefore the immediate medical impact is yet moderate. But only by fundamental research like performed in this ms, we will in the future make medical impact.

Referee #3 (Remarks for Author):

Germann et al. manuscript EMM-2019-10681:
"Neutrophils suppress tumor infiltrating T cells in colon cancer via matrix metalloproteinase mediated activation of TGFβ"

In the revised manuscript, the authors have added new experimentation to address most of the Reviewer's comments and to strengthen the conclusions.

I have concerns about the treatment schedule of the newly added experiments. For instance, in newly added Figure EV2 the authors have combined neutrophil depletion with T cell depletion. These experimental groups are supposed to link with the experiment shown in Figure 3. However, based on the information in the legend of figure EV2, the treatment schedule is very different from that shown in Fig. 3B. In Fig. EV2, the treatment starts 1 day after tamoxifen induction (thus, during the early adenoma phase). Whereas the important message of Figure 3 is that mice with established tumors are being treated. In the text page 8, the authors raise the suggestion that both experiments are performed under the same conditions, which is incorrect. The experiment shown in Fig EV2 should be performed under the same conditions as the experiment in Fig. 3, so the combined depletion and controls (also the T cell depletion only group) should start 8 weeks after tamoxifen induction.

Although mentioned in the figure legend, we apologize that the different treatment schedules between Fig. 3b and Fig EV2 were not explained sufficiently and thus raised the referee's concern. In fact, we conducted experiments in which we strove to deplete CD4+ and CD8+ T cells together with neutrophil reduction in *Apc^{fl/fl}-Cdx2CreERT2* mice according to the treatment schedule shown in Fig. 3B. However, remaining T cells within established colon adenomas were not efficiently depleted (despite efficient T cell depletion in the periphery) while neutrophils were under this regimen.

We therefore chose to co-deplete T cells and neutrophils using a treatment schedule analogous to the T cell depletion scheme described in Figure 1, with which we were able to deplete T cells efficiently.

Neutrophil-only depletion in this manner led to reduced tumor burden (Figure EV2C) compared to control (this is consistent with results achieved under the treatment scheme shown in Figure 3B), while co-depletion of both cell types resulted in increased tumor burden, as expected. The results in EV2C are therefore coherent on their own and moreover, similar to neutrophil depletion in mice with established colon adenomas (as in Figure 3B).

We therefore believe that our statement on page 8 that "When tumor-infiltrating T cells were co-depleted, neutrophil depletion no longer reduced tumor growth" is justified. We once again apologize that the difference between the two treatment schedules employed in Figures 3 and EV2 was not clearly emphasized. Accordingly, we now mention the two different treatment schemes on page 8 as follows:

"In analogy to mice with established colon tumors, treatment of *Apc^{fl/fl}-Cdx2CreERT2* mice with combined anti-Gr1 antibody and CXCR2-inhibitor during and after tumor initiation led to reduced tumor neutrophil infiltration and reduced tumor burden (Fig. EV2). When in this experimental setting tumor-infiltrating T cells were co-depleted, neutrophil depletion no longer reduced tumor growth (Fig. EV2)."

Same concern is true for the experiment in Figure 5D, E. Why did the authors not follow the same treatment schedule as in Figure 3, so treatment of established tumors? In the early adenomas, T cell suppression is less of an issue, as shown in Figure 1, where T cells do limit adenoma formation. These different treatment regimens reduce the transparency of the paper (e.g. due to different treatment schedule, the reader cannot compare the tumor volume of fig. 5E with that of Fig 1C)

The experiment shown in figure 5D,E was performed as a direct reply to referee 1 major point 4 who explicitly requested we treat our experimental animals with TGFbeta inhibitors and MMP9 inhibitors directly after adenoma initiation.

Please see major point of reviewer 1 major point 4 and our reply to it:

Major point 4:

The authors provide strong but exclusive *ex vivo* evidence via a co-culture system that neutrophil-derived MMP9 released by tumor-associated neutrophils activates latent TGF-beta and hence blocks proliferation and cytotoxic activity of T-cells. Neutrophil-derived MMP9 in liver metastatic CRC has been already described recently by Ogawa and colleagues (Clinical Cancer Research, 2019). Importantly, Germann et al. provide important new and mechanistic data on exactly how this might contribute to modulation of the T cell immune-response.

To provide sufficient evidence that this mechanism indeed plays a role during early pre-malignant colon adenoma initiation/formation in vivo, the authors should treat their CDX2-Cre/APC-floxed animals with TGFbeta inhibitors and MMP9 inhibitors directly after adenoma initiation. If their hypothesis is true, then the *de novo* initiated adenoma should get engaged more efficiently by now infiltrating and activated CD8-CTLs, and this should result in a reduced tumor growth/burden.

We thank the reviewer for the very positive comment and the suggested experiment. First, we treated *Apc^{fl/fl}-Cdx2CreERT2* tumor bearing mice with the MMP2/9-inhibitor SB-3CT (MMP2/9i) for

four days to investigate whether this treatment would indeed result in a reduction of TGF β signaling in adenomas. This indeed resulted in a significant decrease in the number and staining intensity of pSMAD3+ cells within tumors (newly added Fig. 5A-C) indicating that the inhibitor lead to reduced TGF β activation. Next, we treated Apc^{fl/fl-Cdx2CreERT2} tumor bearing mice for two weeks (five days on two days off) with either the MMP2/9-inhibitor SB-3CT or TGFBRi (SB431542) and analyzed tumor size oneweek post inhibitor treatment. Both, TGFBRi or MMP2/9i treatment led to reduced average tumor size in Apc^{fl/fl-Cdx2CreERT2} tumor bearing mice. This suggests that targeting Tgfb signaling *in vivo* either via inhibition of its receptor or indirectly via MMP inhibition counteracts early adenoma development. This new data is now shown as Fig. 5D-E and are mentioned on page 10 and 11.

Moreover, please note that according to our data TGF β -mediated immune suppression occurs at the earliest stage of adenoma formation (Fig.EV3C-D, manuscript page 11) and therefore we believe it is prudent to use the inhibitors at an early stage of adenoma development.

The aim of the experiment shown in Figure 5 was to provide proof of concept that inhibition of TGF β signaling through either a TGFBR-inhibitor or a MMP2/9 inhibitor *in vivo* can be at least partially achieved and that this indeed influences tumor development.

References

Pages F, Kirilovsky A, Mlecnik B, Asslaber M, Tosolini M, Bindea G, Lagorce C, Wind P, Marliot F, Bruneval P, Zatloukal K, Trajanoski Z, Berger A, Fridman WH, Galon J (2009) In situ cytotoxic and memory T cells predict outcome in patients with early-stage colorectal cancer. *Journal of clinical oncology : official journal of the American Society of Clinical Oncology* **27**: 5944-5951

Corresponding Author Name: Freddy Radtke

Manuscript Number: EMM-2019-10681-V2